# GIANA allows computationally-efficient TCR clustering and multi-disease repertoire classification by isometric transformation

Hongyi Zhang[1], Xiaowei Zhan[2] & Bo Li [1,3] ✉

Similarity in T-cell receptor (TCR) sequences implies shared antigen specificity between receptors, and could be used to discover novel therapeutic targets. However, existing methods that cluster T-cell receptor sequences by similarity are computationally inefficient, making them impractical to use on the ever-expanding datasets of the immune repertoire. Here, we developed GIANA (Geometric Isometry-based TCR AligNment Algorithm) a computationally efficient tool for this task that provides the same level of clustering specificity as TCRdist at 600 times its speed, and without sacrificing accuracy. GIANA also allows the rapid query of large reference cohorts within minutes. Using GIANA to cluster large-scale TCR datasets provides candidate disease-specific receptors, and provides a new solution to repertoire classification. Querying unseen TCR-seq samples against an existing reference differentiates samples from patients across various cohorts associated with cancer, infectious and autoimmune disease. Our results demonstrate how GIANA could be used as the basis for a TCR-based non-invasive multi-disease diagnostic platform.

[1] Lyda Hill Department of Bioinformatics, UT Southwestern Medical Center, Dallas, TX, USA. [2] Department of Population and Data Science, UT Southwestern Medical Center, Dallas, TX, USA. [3] Department of Immunology, UT Southwestern Medical Center, Dallas, TX, USA. ✉email: bo.li@utsouthwestern.edu

Adaptive immune repertoire is an important regulator of diverse human diseases, and over 10,000 TCR repertoire sequencing (TCR-seq) samples have been generated in recent years. However, interpretation of TCR data has been hindered by the scarcity of known antigen-specificities. Recent studies demonstrated that similarity in the TCR hypervariable complementarity-determining region 3 (CDR3) implicates structural resemblance[1,2] for antigen recognition. Therefore, clustering of similar CDR3s has become an important way to identify antigen-specific receptors.

In the past, a number of TCR clustering methods have been developed to investigate antigen-specific T-cell responses during disease progression or immunotherapy treatments[3–5], such as TCRdist[1], iSMART[4] and GLIPH[2,6]. It is speculated that integrating large number of TCR-seq samples from multiple studies will result in more insights into immune-disease interactions, and create novel opportunities for prognosis and diagnosis[7]. However, methods achieving high-clustering specificity requires pairwise Smith–Waterman (SW) alignment (TCRdist and iSMART) on both the CDR3 sequences and the TCR variable gene (TRBV) alleles, which has quadratic computational complexity that usually cannot scale up to the size of TCR repertoire samples (≥100 K sequences). Motif-based clustering (GLIPH2) achieves higher speed[6], but has much lower specificity[4]. A recent method, clusTCR, applied physiochemical features to numerically encode CDR3 sequences and achieved faster computational speed. This method, however, has omitted the TCR variable gene information, and used the less stringent Hamming distance to replace the SW alignment, which resulted in lower clustering purity[8]. Therefore, none of the existing TCR clustering methods are suitable to analyze large cohorts of TCR-seq samples.

To address this challenge, we introduced Geometric Isometry-based TCR AligNment Algorithm (GIANA), a mathematical framework to transform the CDR3 sequences, which converted the sequence alignment and clustering problem into a classic nearest neighbor search in the high-dimensional Euclidean space. This transformation significantly improved the computational efficiency for TCR pairwise comparison and scaled up to $10^6$ to $10^7$ sequences. In this work, we demonstrated that by pooling thousands of TCR repertoire samples, GIANA can identify novel disease-associated TCRs and assign unseen samples into the correct disease categories. Thus, our approach might open an alternative avenue for an immune-based multi-disease diagnostic platform.

## Results

**An isometric embedding framework for ultra-fast TCR alignment.** GIANA began with an approximated solution to the isometric embedding of BLOSUM62 matrix using multidimensional scaling (MDS) (Fig. S1), which generated a numeric vector for each of the 20 amino acids. Subsequently, CDR3 strings were modeled as serial non-commuting linear transformations on the MDS vectors and were represented as coordinates in the high-dimensional space. The unitary transformation matrix was an element of the 6-order cyclic group ($G_6$), which produced near-perfect linear correlations between the Euclidean distances of a pair of strings and their Smith–Waterman alignment scores (Fig. S2). At default, isometric distance cutoff (-t) of 10, all TCR pairs with high Smith–Waterman alignment scores were included in the downstream clusters. Fast, index-based nearest neighbor search and recursive centroid grouping were then performed on the coordinates to identify CDR3 pre-clusters, which were subsequently filtered for matched TRBV alleles and high alignment scores using a k-mer guided search table, to produce the final TCR clusters as output (Fig. 1).

In parallel to $G_6$ transformation, we also implemented a naïve method with stacked MDS vectors as the coordinates of the input CDR3 strings (GIANAsv), similar to a recent work[8] (Fig. S3). We developed a benchmark with 10 K, 20 K, …, 100 K TCRs from a healthy donor's TCR-seq sample[9] and applied five competing methods, GIANA, GIANAsv, iSMART[4], TCRdist[1] and GLIPH2[6] for speed and memory tests (Fig. 2a and Fig. S4). We excluded a similar method, ALICE[10], as it was optimized to find the most variable clones in longitudinal data only. GIANA has the lowest time cost throughout the benchmark, taking 23.9 s to process 100 K sequences, whereas TCRdist took 14,338 s. GIANAsv is slower than GIANA by a factor of 2.2 (Table S1). This is expected because stacked vector encoding resulted in higher dimensionality of the isometric embedding space, and increased the time cost during nearest neighbor search. Notably, GLIPH2 is the fastest algorithm besides GIANA/GIANAsv, because it avoided pairwise alignment through motif-guided search.

**GIANA achieves higher accuracy in predicting antigen-specific TCRs.** As antigen-specificity is the most desirable feature of TCR clustering, we next evaluated the clustering accuracy of all the methods. GIANAsv was excluded from this analysis because it has

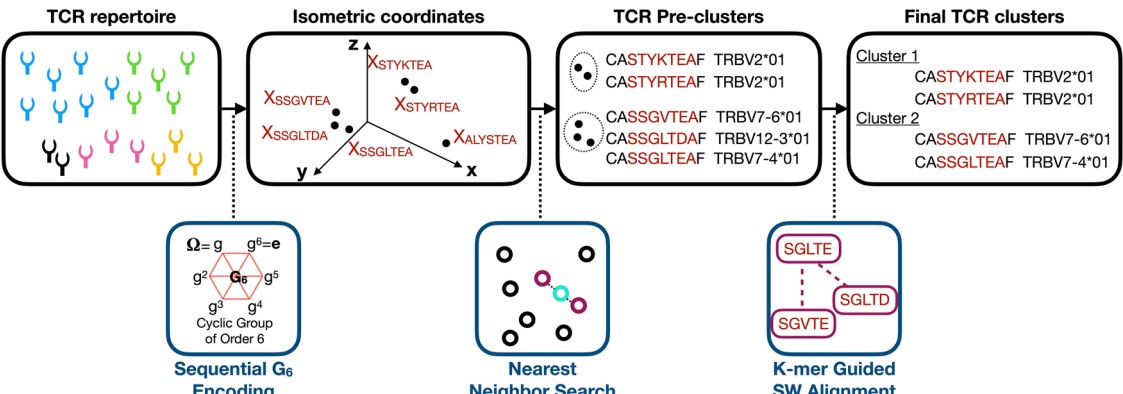

**Fig. 1 Schematic illustration of GIANA.** GIANA workflow. GIANA began with encoding of short CDR3 peptide sequences into numeric vectors through a sequence of unitary transformations. The transformation involves an element of 6th order cyclic group. After encoding, each CDR3 sequence was projected to high-dimensional Euclidean space, allowing fast nearest neighbor search for clustering. Follow-up filtering steps were performed to match the TRBV gene alleles and remove pairs with low alignment scores.

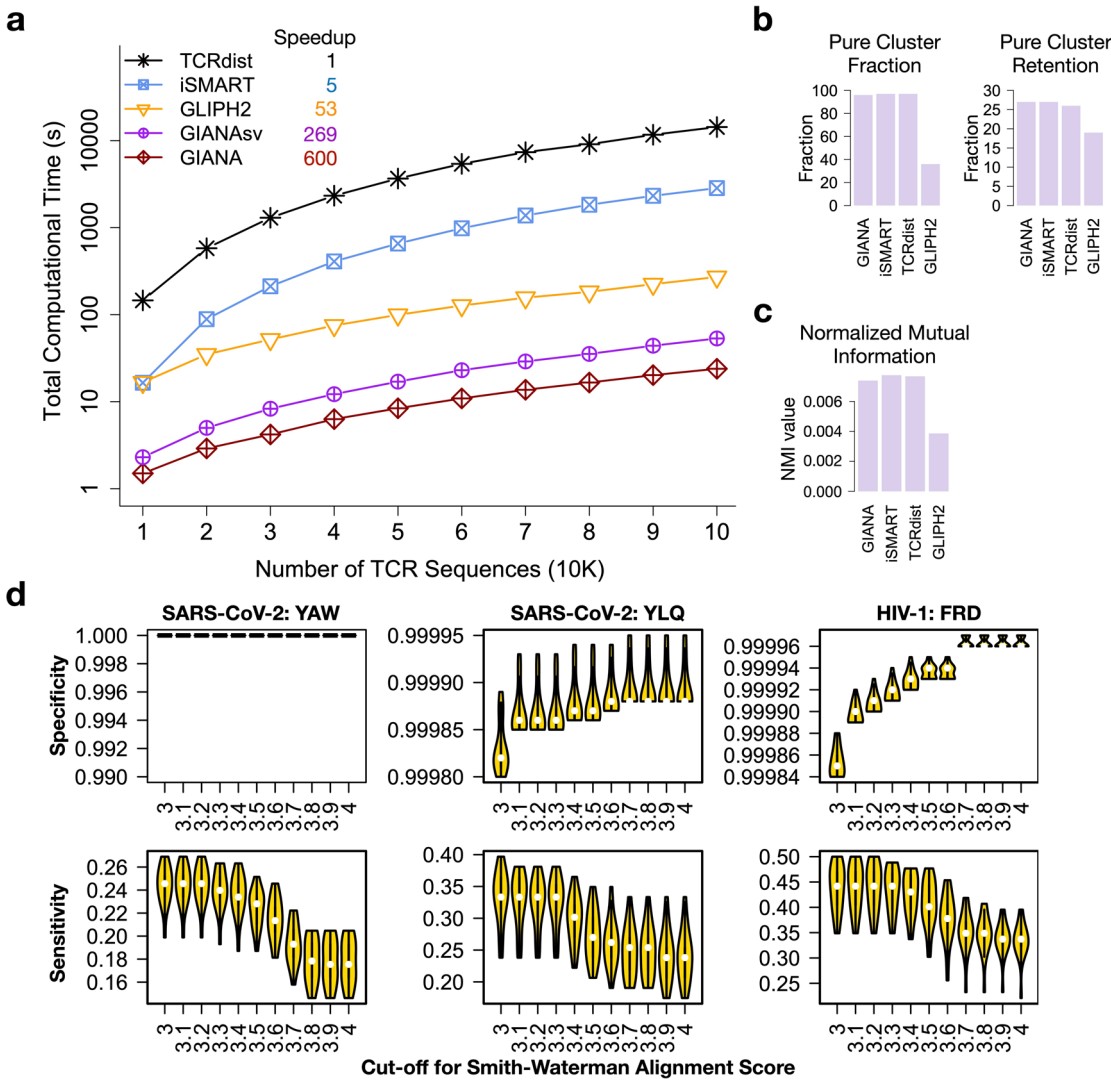

**Fig. 2 Performance evaluation and comparison of different methods. a** Comparison of time complexity for five competing TCR clustering algorithms. Speedup was calculated based on the time cost for the 100 K TCR sample. **b** Comparison of purity cluster fraction and retention of four methods. y-axis shows the precision or sensitivity in percentage. **c** Normalized mutual information comparison of the four methods. **d** Sensitivity and Specificity of GIANA when applied to large and noisy TCR-seq samples. x-axis is the cutoff for Smith–Waterman alignment score, a key parameter in GIANA, with maximum 4.0. Higher cutoff results in higher specificity at the cost of reduced sensitivity.

theoretically identical performance as GIANA except for computational efficiency. We collected 61,366 non-redundant known TCR/antigen pairs from the public domain[1,11–13], covering over 900 different epitopes from diverse pathogens. We first defined cluster purity as previously described:[14] the percentage of TCRs specific to the most common epitope in a given cluster. A "pure cluster" is defined to have a purity equal to 1. We defined "Pure cluster fraction" as the percentage of pure clusters in the output (Fig. 2b and Table S2). The fractions for GIANA (96%), iSMART (97%), and TCRdist (97%) are similar, yet substantially lower for GLIPH2 (35%). Pure cluster retention was defined as the total number of TCRs in all the pure clusters divided by the number of all the testing TCRs, and we confirmed that GIANA also has similar level of retention (27%) as other methods, except for GLIPH2 (19%). For the three methods that rely on Smith–Waterman alignment (GIANA, iSMART and TCRdist), we further explored the impact of a range of alignment score cutoffs (-S option in GIANA). We observed similar performances for the three methods at stringent cutoffs (Fig. S5).

Pure cluster fraction and retention emphasized the clusters with 100% purity, which are more frequently seen in smaller

clusters. Methods that produced larger clusters may have worse performance due to impure clustering. Therefore, we also measured normalized mutual information (NMI) between TCR clusters and epitope specificity, using the same training dataset. We observed similar levels of NMI across all the methods using Smith–Waterman alignment, and GLIPH2 remained to be the lowest (Fig. 2c).

Next, we investigated if GIANA is able to retrieve antigen-specific TCRs from real, large, and noisy TCR-seq samples, using TCRs with known antigen-specificity. From the above benchmark antigen-specific TCRs, we selected those specific to three epitopes expected to be missing in healthy individuals: the YAW and YLQ epitopes[15] from the recent outbreak of SARS-CoV-2 virus and the FRD epitope[16] from HIV-1 virus. 20% of these TCRs were mixed with 100,000 TCRs from a healthy donor as testing data, and we used the remaining non-overlapping 80% antigen-specific TCRs as training data to recover the test sequences. Any sequence clustered with the training data will be called as "positive." True positives are the 20% spiked-in antigen-specific TCRs, whereas false positives are those from the healthy donor. For all three

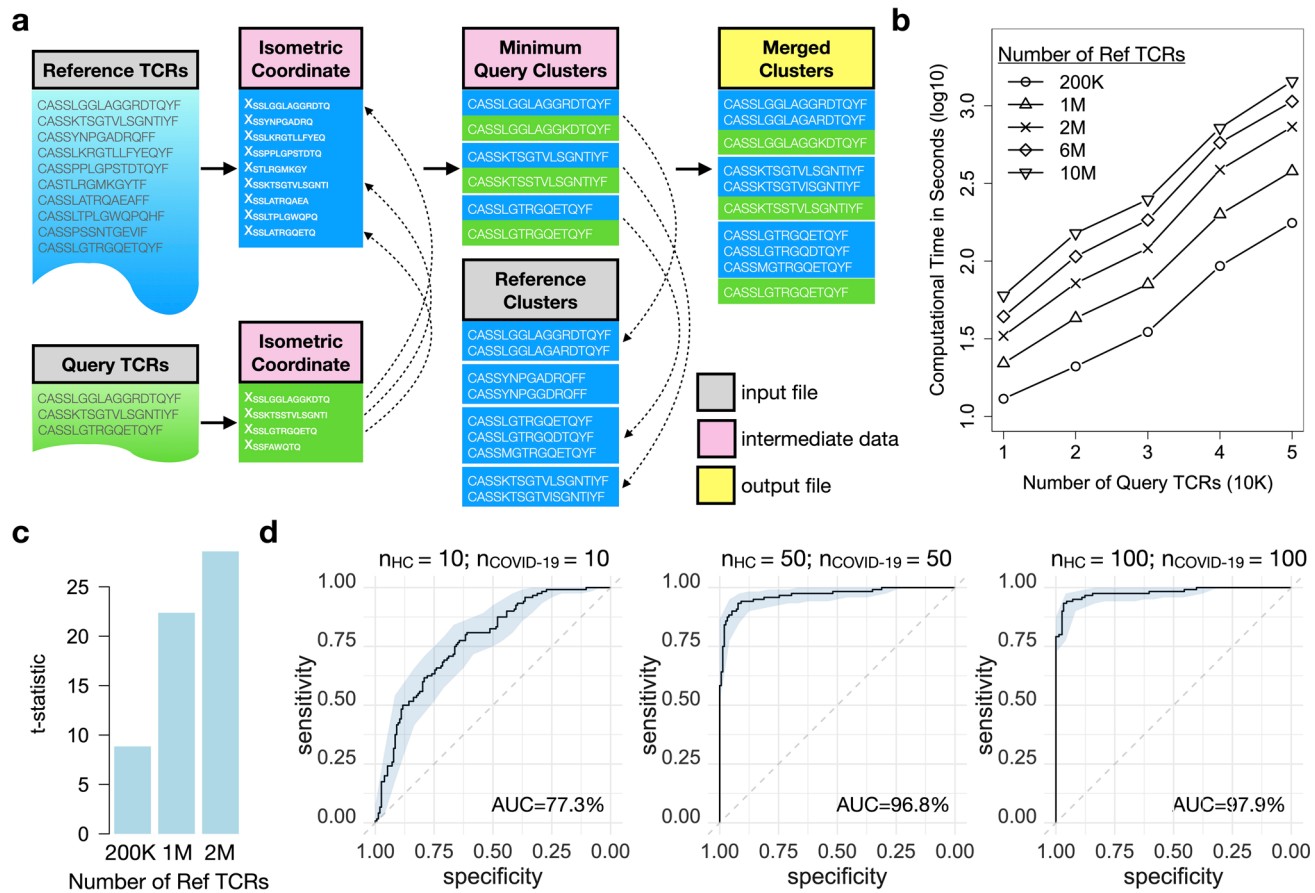

**Fig. 3 Fast query of previously generated reference TCR clustering data for accurate repertoire classification. a** Illustration of fast GIANA query based on isometric transformation. GIANA performs nearest neighbor search of each query TCR against the reference coordinates, selects closely located neighbors in the Euclidean space, and merges with reference TCR clusters. Dashed arrows implicate search directions. **b** Time complexity evaluation of GIANA query module using reference/query data with different number of TCRs. **c** Degree of separation of query COVID-19 patients from healthy controls by clustering against the reference datasets. The number of TCRs of the reference data was shown as *x*-axis labels. Two sample *t*-test was performed using the COVID-19 fractions estimated from the query data to obtain the t-statistics. All *p*-values were significant at the level of $2.2 \times 10^{-16}$. **d** ROC curves using COVID-19 fraction as the single predictor. The numbers of COVID-19 and HC samples were labeled in figure title, with each sample containing 10 K TCR sequences. Shaded area labeled the 95% confidence interval of the ROC curves, which were estimated from 2,000 stratified bootstraps.

epitopes, GIANA achieved over 99.99% specificity at 20–50% sensitivity (Fig. 2d). Although GLIPH2 reached higher sensitivity, its specificity is lower than GIANA (Fig. S6a). More importantly, the positive prediction values (PPV) of GIANA reached over 60% for all epitopes, while the PPVs of GLIPH2 for 2 out of the 3 epitopes were lower than 20% (Fig. S6b).

**Ultra-fast sample query and TCR repertoire classification.** The high speed and specificity of GIANA motivated us to further develop a query module to cluster new TCR samples with an existing reference dataset, a function that is missing in all current tools. We transformed reference and query TCRs into Euclidean space with linear complexity and searched for the nearest neighbors of each query sequence, which were processed into TCR clusters and merged with the reference data (Fig. 3a). We evaluated the speed of this approach using query and reference datasets with different numbers of TCRs (Fig. 3b). As expected, GIANA is computationally efficient as demonstrated by the fact that it took 176 s to query $10^4$ TCRs against $10^7$ reference sequences (Table S3).

Repertoire classification is an important task with immediate applications to disease diagnosis and prognosis[7]. In the past, this task has been approached by multiple instance learning[17] or deep learning[18]. We next explored if GIANA query can also be used to

classify TCR repertoires. First, we generated 3 reference datasets with 20, 100 or 200 TCR-seq samples, evenly split into COVID-19 patients and healthy controls (HC). An additional 154 COVID-19 and 120 HC samples were queried to each of the references. For each query sample, we calculated the fraction of TCRs co-clustered with COVID-19 reference patients. Interestingly, this fraction is significantly higher for the COVID-19 patients in the query samples, with increasing separation from query HCs as the size of reference data increases (Fig. 3c and Fig. S7a). Using this fraction as a predictor, we observed increasing Area Under the receiver operation Curve (AUC) for reference with larger sample size (Fig. 3d). Notably, with 2 million reference TCRs, the sensitivity (79%) and specificity (100%) surpassed some existing test for COVID-19[19], suggesting the potential utilities of this approach in disease diagnosis. The fact that the accuracy of repertoire classification improved with more reference samples is likely due to that disease-specific TCRs are usually shared at low frequencies[20]. Consequently, a larger reference data will have higher clustering probability, smaller dispersion and better precision. Indeed, we observed decreasing coefficient of variance of COVID-19 fractions with more reference samples (Fig. S7b).

We extended the above efforts by building a large reference dataset containing 10 million TCRs, which consisted of

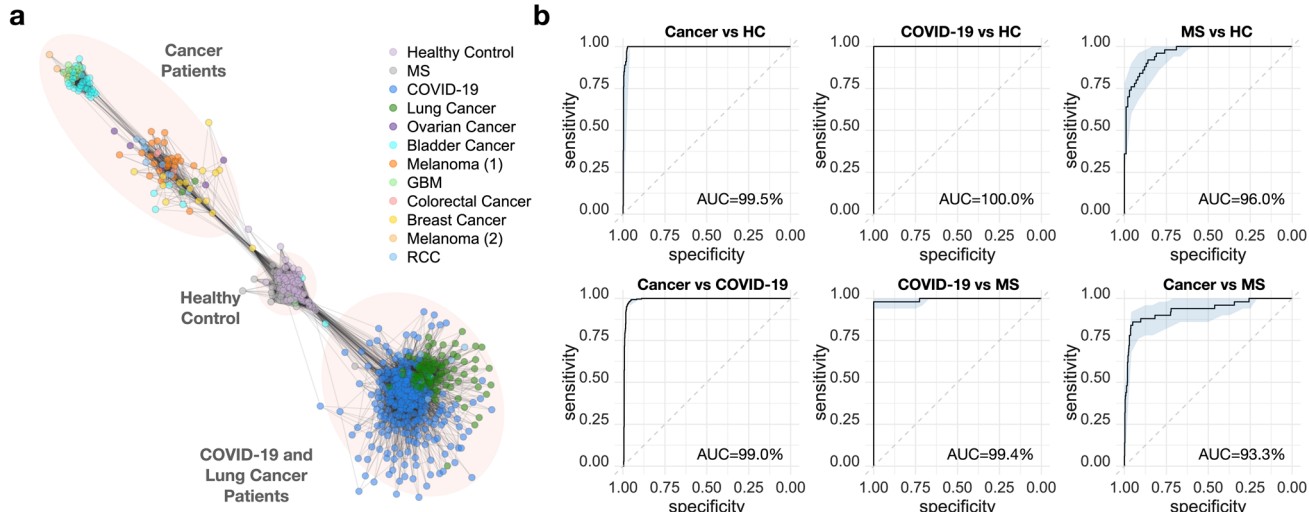

**Fig. 4 Disease-specific grouping of TCR repertoire samples via ultra-large-scale clustering. a** Graphic representation for the similarities of the TCR-seq samples based on TCR co-clustering. Sample-wise count-sharing matrix was computed from the original TCR clustering results of the 1,213 reference samples. Spearman correlation matrix was calculated based on counts of co-clustered TCRs, with pairs having a correlation value ≤0.4 set to be zero. The resulting sparse matrix was used to generate the graph. Nodes with fewer than two connections were removed to visualize the sample groups. **b** ROC curves using disease fractions calculated from co-clustered TCRs. AUC values were labeled at the bottom right of each panel. 95% confidence intervals were calculated using 2,000 stratified bootstraps. Disease abbreviations: GBM for glioblastoma multiforme; RCC for renal clear cell carcinoma; MS for multiple sclerosis.

1,213 samples from cancer, COVID-19 and multiple-sclerosis (MS) patients and HCs[9,21–34] (Table S4). First, we applied GIANA to perform antigen-specific clustering of all 10M TCRs and investigated the similarity of different repertoire samples measured by the level of shared TCR clusters. We observed clear separations of most cancer patients from healthy donors and MS patients. Interestingly, lung cancer and COVID-19 patients together formed a separate cluster (Fig. 4a). It is known that local inflammatory conditions, such as viral infection or cancer, could release tissue-resident T cells into the circulation[35], a likely cause for TCR repertoire sharing. Our findings further suggested that in the lung tissue, the magnitude of T cell egress might be high enough to transcend disease types. However, currently it is not feasible to experimentally validate this observation due to the lack of paired α chain information.

**A multi-disease detection platform through ultra-large-scale TCR clustering and query.** The ultra-large-scale clustering by GIANA also allowed us to inspect disease-specific vs. tissue-specific TCRs. We divided the TCR clusters in the lung cancer and COVID-19 patients into three categories: (i) COVID-19 specific; (ii) lung cancer specific; (iii) shared between the two diseases. We observed significantly higher clonal frequency of category (i) vs. (iii) for COVID-19 patients, whereas there is no difference between category (ii) and (iii) for the lung cancer patients (Fig. S8a). TCR frequencies were matched within same cohort to avoid batch effect, and thus, the higher abundance of COVID-19-specific TCRs is likely caused by an immune response to SARS-CoV-2. Indeed, only COVID-19-specific TCRs underwent dynamic regulation after viral infection, which peaked within the first 2 weeks post-exposure and decreased afterwards. In contrast, clonal abundance of shared TCRs were unaffected by the timeline after SARS-CoV-2 infection (Fig. S8b). In conclusion, clustering on large TCR repertoire samples might reveal shared disease-specific TCRs, which may provide a finer solution to repertoire classification.

Therefore, we tested if clustered TCRs can be used as markers to assign repertoire samples into multiple diseases, by

implementing a leave-one-out validation approach. Specifically, for a given sample, we calculated the fractions of TCRs co-clustered with cancer, COVID-19, MS patients or healthy controls in the reference cohort, excluding the sample itself. This method yielded four class fractions for each sample, which added up to 1. We used the HC fraction to separate patients from healthy donors and observed near perfect accuracies for all three diseases (Fig. 4b). To differentiate a pair of diseases, we used the differences between the two corresponding fractions as the predictor, which also led to high (≥93%) AUC values. The ability to distinguish lung cancer from COVID-19 was not contradictory to the apparent grouping of the two diseases (Fig. 4a) because within-disease similarity was still higher (Fig. S9). However, as most of the diseases were derived from only one study, it raised the concern that the predictability might be contributed by unknown cohort-specific batch effects.

To test out this possibility, we investigated if GIANA can predict the disease labels of unseen samples from independent cohorts. We applied GIANA to query 267 new TCR-seq samples of the three disease categories and 153 HC samples[9,18,25,36–39] (Table S5) against the reference dataset. All samples were derived from peripheral blood. We used the same approach to calculate the fractions of TCRs co-clustered with reference cancer, COVID-19, MS, or HC sequences. Without any model fitting, this simple approach can already distinguish each sample category from the others (Fig. 5a). HC fractions distinguish all 3 diseases at over 91% accuracy, whereas pairwise separation between diseases all reached above 87% AUCs (Fig. 5b). Since the query samples were derived from studies not included in the reference data, the high AUCs were unlikely caused by unknown batch or cohort-specific effects and thus could reflect the real predictability for the three types of diseases.

## Discussion

In summary, GIANA is a fast TCR clustering algorithm that efficiently handles tens of millions of sequences. It achieved the same level of accuracy as the best existing methods and was able to retrieve TCRs specific to known antigens with high accuracy.

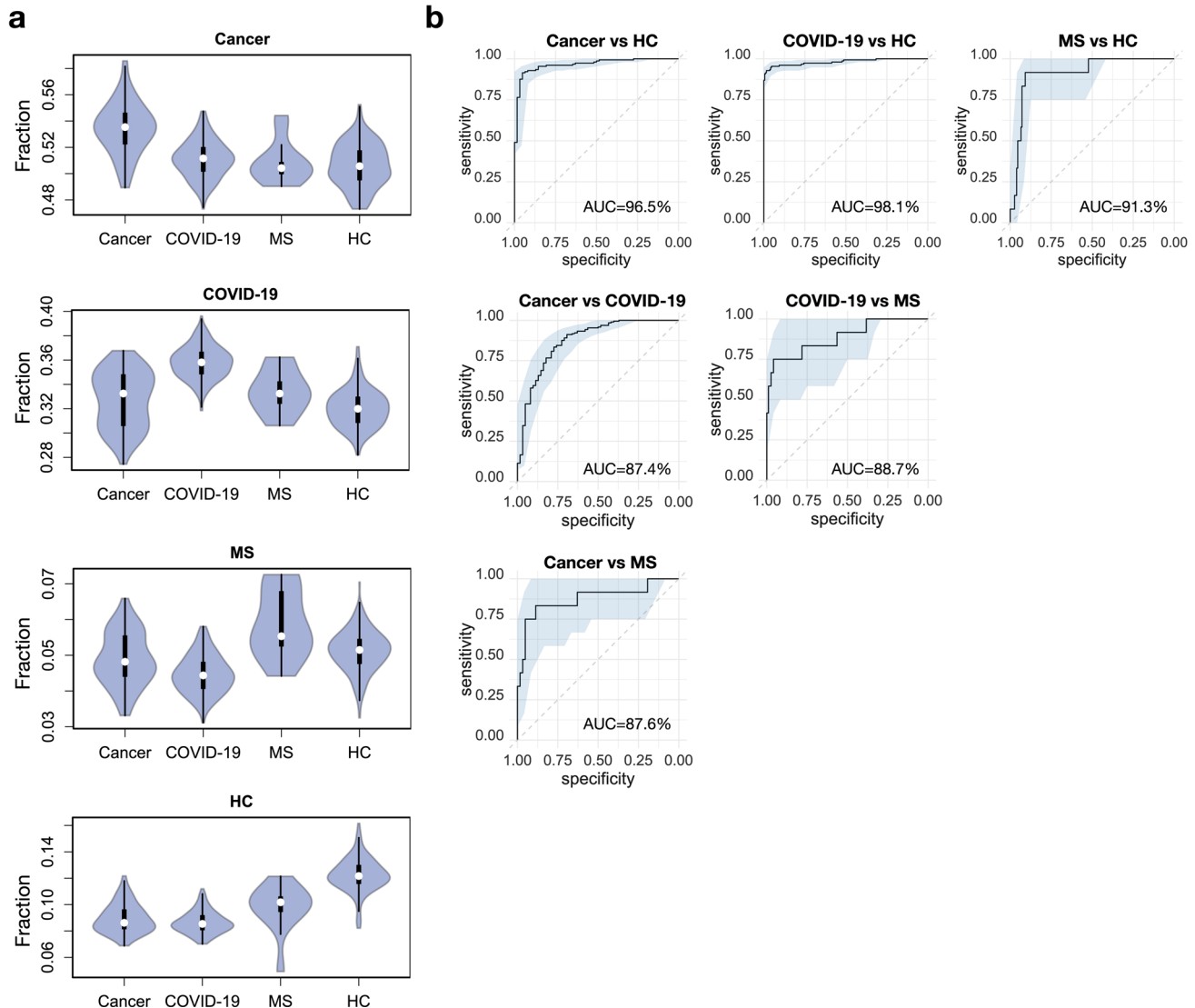

**Fig. 5 Reference-based multiple disease classification of unseen TCR repertoire samples. a** Violin plot showing the distribution of class fractions of cancer, COVID-19, multiple sclerosis (MS) patients and healthy controls (HC). Cancer fraction was calculated as the proportion of query TCRs clustered with reference TCRs from the cancer patients. Other class fractions were defined in the same way. Sample size: HC: $n = 153$, COVID-19: $n = 193$, Cancer: $n = 62$, MS: $n = 6$. **b** ROC curves using disease class fractions as single predictor for pairwise separation of the four disease classes. The fraction was the percentage of TCRs co-clustered with a given class of samples in the reference dataset. AUC values were labeled at the bottom right of each panel. 95% confidence intervals were calculated using 2,000 stratified bootstraps.

The ultra-large-scale TCR clustering and fast query of novel samples also enabled reference-based repertoire classification. To date, GIANA can also analyze single cell RNA-seq data with TCR regions solved, and it is possible to query TCRs from the scRNA-seq data against the large database of TCR repertoire samples in the public domain to gain new insights over shared antigen-specificity. With minimum modifications, GIANA is applicable to cluster or query B-cell receptor sequencing data as well. Furthermore, the mathematical framework to perform isometric embedding may provide an alternative solution to the classic short DNA or protein sequence alignment problems in the future.

Unsupervised TCR clustering is a fundamental analysis of immune repertoire data. In the ideal scenario, all TCRs specific to the same epitope should be included in the same cluster. However, this is not feasible for sequence similarity or motif-based clustering approach, due to the putative diversity in TCR sequences of shared specificity[1]. Such diversity is caused by the distinct docking strategies of T-cell receptors. For example, TCRs

specific to the influenza GIL epitope usually contain the classic RSS/RSA motif in the CDR3 region, yet a related study[40] reported that LGGW motif also elicits strong binding to GIL from a different direction. Such structural variation cannot be captured by simple Smith–Waterman alignment or motif grouping. Consequently, CDR3s with dissimilar motifs will be fragmented into smaller clusters despite their shared specificity, which is a common limitation to the current methods.

There are several limitations in our study: first, unlike GLIPH2, HLA alleles were not considered in GIANA, as such data is unavailable in most current studies. With HLA typing included, the accuracy of TCR clustering and query methods is expected to improve. Second, GIANA does not support gap alignment, as the current isometric encoding framework only applies to sequences with the same length. However, it has similar level of clustering accuracy as the methods that consider gaps, such as TCRdist. This is because allowing gaps will reduce clustering specificity and compromise the prediction accuracy[4]. Third, in this work, we

simply used the TCR fractions to assign disease classes. With more data, this effort can be improved by machine learning models to optimize the prediction accuracy. Fourth, we compared all cancer patients with other diseases together but were not able to differentiate cancer localizations. We anticipate the power to separate cancer types with enough relevant TCR-seq samples as the reference. Finally, although the current method already achieved high accuracy of repertoire classification, the diagnostic value of this platform requires further validation with prospectively collected patient samples.

As demonstrated in autoimmune and infectious diseases, antigen-specific public TCRs shared at low frequencies are potentially important biomarkers[20,41,42], which can be detected by comparing large amount of TCRs from thousands of individuals. Methods have been developed to individually detect cancer[17,18], COVID-19[20], or multiple sclerosis[43] using immune repertoire, but none has been able to simultaneously diagnose and separate different diseases. In contrast, our effort could be developed into a unified platform to diagnose infectious disease, autoimmune disorders and cancer. Such a platform has been proposed in the literature[7], and in this work, we provided a prototype to achieve this goal.

We believe this is potentially a significant advance because: first, disease diagnosis is mainly symptom-driven for decades, with each disease requiring a distinct set of signatures obtained from diverse clinical assays, such as radioactive imaging, liquid biopsy, invasive endoscopy, surgery, etc. The feasibility of using the immune system as a single biomarker to indicate multiple diseases could shift the paradigm from symptom-driven to immune-response-driven, which provides a universal solution to many immune-related disorders. Additionally, differential diagnosis is usually a clinical challenge, and adding more diseases to the platform will further reduce the diagnostic specificity. We provided a solution to this problem by showing that the platform can increase its prediction accuracy by the inclusion of more TCR-seq samples. Further, as immune responses are usually ahead of any measurable symptoms, this platform has the potential to detect diseases at its early stages, where most diseases are curable or easy to manage. We have already demonstrated this fact for cancer diagnosis[18], and the principle of immune regulation also applies to autoimmune disorders, such as multiple sclerosis. Finally, since this platform only requires a small amount of blood to perform targeted V(D)J capture, it can serve as a non-invasive test at low cost. Together, we anticipate GIANA to be widely used to find antigen-specific TCR clusters, to retrieve sequences specific to known pathogens, such as SARS-CoV-2, and to facilitate disease diagnosis with the fast-growing body of TCR data in cancer, immunology and clinical studies.

## Methods

**TCR-seq data collection.** All TCR repertoire sequencing samples were accessed via immuneACCESS of Adaptive Biotechnology, which currently hosts the largest database of TCR-seq samples, all profiled using the immunoSEQ platform. Antigen-specific TCR and the matching antigens were pooled from the VDJdb[12], the Immune Epitope Database and Analysis Resource (IEDB), and previous literature[1,2]. TCRs specific to more than one epitope were removed to avoid conflicts.

**GIANA method description**

*Mathematical framework for isometric embedding of CDR3 sequences.* The goal is to find a numeric representation (also the coordinates in high-dimensional space) $x$ of any short peptide sequence $s$, such that, for $s_i$ and $s_j$, the Euclidean distance between the two coordinates $x_i$ and $x_j$: $\|x_i - x_j\|$, is perfectly correlated to the sequence similarity score measured by putative evolutionary substitution matrix. We refer this problem as "isometric embedding of short sequences". This concept is introduced to solve the numeric encoding problem of the CDR3 sequences, typically with lengths ranging from 12 to 17 amino acids. In this section, we presented the process of finding the mathematical transformation of a given CDR3 sequence

that approximately satisfied isometry. First, we found an approximately isometric embedding for the BLOSUM62 matrix. The problem is defined as below:

Let amino acid $A_i$ be represented by $\beta_i$, a numeric vector in real space $\mathbb{R}^r$. The dimension of the real space $r$ is determined by the rank of the Euclidean Distance Matrix, or EDM. In this scenario, let $M$ denote the dissimilarity matrix derived by BLOSUM62: $M = 4 - \text{BLOSUM62}$, and the diagonal values of $M$ are set 0. Isometry indicates that

$$\|\beta_i - \beta_j\| = M_{ij} \tag{1}$$

In 1974, C. L. Morgan, in his work "*Embedding Metric Spaces in Euclidean Space*"[44] proved that the solution to this problem exists if, and only if the EDM is flat, and the embedding space has dimension no greater than $n$, where $n$ is the dimension of EDM. Unfortunately, BLOSUM62 matrix is not even an EDM, since it does not satisfy the triangular rule:

$$M_{ik} + M_{kj} \geq M_{ij}, for \forall i, j, k \tag{2}$$

Therefore, there does not exist an exact isometric embedding of BLOSUM62. However, Multidimensional Scaling (MDS) provides an approximate solution, which applies to the cases where $M$ is not an EDM. We used MDS to derive the embedding vectors $\beta_i$. The transformed distance matrix $M$ has rank $r = 13$. With classic MDS, the maximum dimension for the embedding space is 13. We applied the non-metric MDS[45] calculation using the sklearn package in Python and could explore dimensionality higher than 13. To maximize the embedding isometry, we first selected 2,300 training TCRs of length 14 from a previously described TCGA dataset[4] and calculated the pairwise SW alignment scores. We applied the MDS to obtain isometric embedding vectors of different dimensions, ranging from 13 to 19. For each length, we calculated the Euclidean coordinates of the CDR3 sequences as described in the GIANA method and compared the pairwise distance with SW scores. The maximum score was observed with dimension 16 (Spearman's $\rho = -0.973$, Figure S1). This representation achieves 87% similarity to the BLOSUM matrix:

$$\|\beta_i - \beta_j\| \approx M_{ij} \tag{3}$$

Next, we introduce a numeric encoding scheme, such that each amino acid is considered as an "operator", parable to the concept in quantum physics. In general, operator $\mathbf{A}$ is a mathematical transformation to an existing wave function $\phi$. The operation can be applied to wave functions denoted by the Dirac bracket: $\mathbf{A}|\phi>$. One example is the angular moment operators $\mathbf{L}_X$, $\mathbf{L}_y$, $\mathbf{L}_z$. Here, we define the operator for amino acid $i$ to be $\mathbf{A}_i$, which applies to a numeric vector x in the following way:

$$\mathbf{A}_i x := \boldsymbol{\Omega} \times (x + \beta_i) \tag{4}$$

Here $\boldsymbol{\Omega}$ is a matrix that needs to be determined. This definition emphasizes the ordering of letters in a sequence as operators are non-commuting: $\mathbf{A}_i\mathbf{A}_j \neq \mathbf{A}_j\mathbf{A}_i$ if $i \neq j$. A CDR3 sequence is then serial linear operations on some initial vector, $\beta_0$. To simply calculation, we always let $\beta_0 = 0$. Therefore, after the operation on the rightmost amino acid, the coordinates become:

$$\mathbf{A}_i\beta_0 := \boldsymbol{\Omega} \times (\beta_i + 0) = \boldsymbol{\Omega} \times \beta_i \tag{5}$$

Below we use several cases to illustrate the desirable qualities of $\boldsymbol{\Omega}$.
*Case 1*: Single Mismatch
We begin with the simple case where two amino acid sequences are off by one amino acid. For example, sequence $s_1 = \mathbf{A}_k\mathbf{A}_i$, and sequence $s_2 = \mathbf{A}_k\mathbf{A}_j$. Their numeric encoding vectors can be calculated as below:

$$s_1 = \mathbf{A}_k\mathbf{A}_i$$

$$s_2 = \mathbf{A}_k\mathbf{A}_j$$

$$x_1 = \boldsymbol{\Omega} \times (\boldsymbol{\Omega} \times \beta_i + \beta_k)$$

$$x_2 = \boldsymbol{\Omega} \times (\boldsymbol{\Omega} \times \beta_j + \beta_k)$$

$x_1$ and $x_2$ are the encoding vectors of $s_1$ and $s_2$. The Euclidean distance between $s_1$ and $s_2$ can be calculated by:

$$\begin{aligned} \|x_1 - x_2\| &= \delta^T\delta, where \delta = x_1 - x_2 \\ &= (\boldsymbol{\Omega}^2(\beta_i - \beta_j))^T(\boldsymbol{\Omega}^2(\beta_i - \beta_j)) \\ &= (\beta_i - \beta_j)^T\boldsymbol{\Omega}^T\boldsymbol{\Omega}^T\boldsymbol{\Omega}\boldsymbol{\Omega}(\beta_i - \beta_j) \end{aligned} \tag{6}$$

We hope the above value is equal to the distance between amino acid $A_i$ and amino acid $A_j$, as this is the only mismatch. Thus:

$$(\beta_i - \beta_j)^T\boldsymbol{\Omega}^T\boldsymbol{\Omega}^T\boldsymbol{\Omega}\boldsymbol{\Omega}(\beta_i - \beta_j) = (\beta_i - \beta_j)^T(\beta_i - \beta_j) \approx M_{ij}$$

Without losing generality, we let $\boldsymbol{\Omega}^T\boldsymbol{\Omega} = \mathbf{I}$, i.e. $\boldsymbol{\Omega}$ is a unitary matrix. It can be easily shown that longer sequences with one mismatch follow the same argument above.
*Case 2*: Two Consecutive Mismatches

Now we define $s_1 = A_i A_j$ and $s_2 = A_t A_k$. It can be shown that the distance between the embedding vectors $x_1$ and $x_2$ is:

$$\|x_1 - x_2\| = (\Omega^2(\beta_j - \beta_k) + \Omega(\beta_i - \beta_t))^T(\Omega^2(\beta_j - \beta_k) + \Omega(\beta_i - \beta_t))$$
$$= (\beta_j - \beta_k)^T(\beta_j - \beta_k) + (\beta_i - \beta_t)^T(\beta_i - \beta_t) - 2(\beta_i - \beta_t)^T\Omega(\beta_j - \beta_k) \quad (7)$$
$$\approx M_{jk} + M_{it} - 2(\beta_i - \beta_t)^T\Omega(\beta_j - \beta_k)$$

Ideally, we want the third term to be zero for $\forall i, j, t, k$. One solution is to let $\Omega$ be a rotation in $\mathbb{R}^{2r}$, by imposing a perpendicular rotation from the first $r$ dimensional space to the complement space. A simple realization is:

$$\Omega_2 = \begin{pmatrix} \underline{0} & I \\ I & \underline{0} \end{pmatrix} \quad (8)$$

where $I$ is the $r$-dimensional identity matrix, and $\underline{0}$ is an $r$-dimension zero matrix. In fact, the $\Omega$ defined this way is a representation of order 2 cyclic group $G_2$. $G_2$ has only two elements: $e$ and $g$, with $g^2 = e$. This notation is useful when we make extension to the scenario with multiple consecutive mismatches. The $\beta_i$ will be extended to $\mathbb{R}^{2r}$ accordingly, with the first $r$ dimensions filled with the values derived from the MDS embedding and the remaining dimensions filled with a vector of zeros:

$$\beta_i^{2r} = \begin{pmatrix} \beta_i \\ 0 \end{pmatrix} \quad (9)$$

Here, 0 is a vector of zeros with dimension $r$. The new vector satisfies:

$$\beta_i^{2r^T} \Omega_2 \beta_j^{2r} = (\beta_i^T, \mathbf{0}^T)\begin{pmatrix} \underline{0} & I \\ I & \underline{0} \end{pmatrix}\begin{pmatrix} \beta_j \\ 0 \end{pmatrix} = (\beta_i^T, \mathbf{0}^T)\begin{pmatrix} 0 \\ \beta_j \end{pmatrix} = 0$$

*Case 3*: Multiple Consecutive Mismatches

It can be proven that for a sequence $s_i = A_{i_k}A_{i_{k-1}} \ldots A_{i_1}, k \geq 3$, its encoding vector can be written as:

$$x_i = \sum_{l=1}^{k} \Omega^l \beta_{i_l}, l = 1, 2, \ldots, k$$

Let's consider another sequence $s_j = A_{j_k}A_{j_{k-1}} \ldots A_{j_1}$:

$$x_j = \sum_{l=1}^{k} \Omega^l \beta_{j_l}, l = 1, 2, \ldots, k$$

The distance of $x_i$ and $x_j$ is:

$$\|x_i - x_j\| = \sum_{l=1}^{k}(\beta_{i_l} - \beta_{j_l})^T(\beta_{i_l} - \beta_{j_l}) - 2\sum_{u=2}^{k}\sum_{1 \leq v < u}^{K}(\beta_{i_u} - \beta_{j_u})^T(\Omega^u)^T(\Omega^v)(\beta_{i_v} - \beta_{j_v}) \quad (10)$$

In the ideal scenario, all the terms in the double $\sum$ are 0, and the distance between $s_i$ and $s_j$ is simply $\sum_{l=1}^{k} M_{i_j l_j}$. This requires:

$$(\beta_{i_u} - \beta_{j_u})^T \Omega^{u-v}(\beta_{i_v} - \beta_{j_v}) \quad (11)$$

is 0 for $\forall u, v, k$. Or, generally: $x^T \Omega^{u-v}y = 0$ for $\forall x, y \in \mathbb{R}^r$. There is no solution for such an $\Omega$ in $\mathbb{R}^r$, but similar to *Case 2*, we may increase the dimensionality of the embedding space to $kr$. In this way, one can always construct $\Omega$ from $n$ order cyclic group $G_n$, which is an Abel group. However, increased dimensionality will increase the computational complexity in the encoding step by a factor of $O(k)$. Also, even with the exact solution, the distance calculated by MDS embedding does not perfectly align with BLOSUM62 scores. Therefore, there is a trade-off to increase dimensionality. In practice, we set $k$ to be 6. This is a reasonable number considering the median length of T cell CDR3 sequences is 14, with the first 4 and last 1 amino acids almost invariant. To construct the corresponding matrix, we first derive a representation of the element in $G_3$ by:

$$\Omega_3 = \begin{pmatrix} \underline{0} & \underline{0} & I \\ I & \underline{0} & \underline{0} \\ \underline{0} & I & \underline{0} \end{pmatrix} \quad (12)$$

Both $G_2$ and $G_3$ are normal subgroups of $G_6$, with $G_6 = G_2 \otimes G_3$. Therefore, from $\Omega_2$ and $\Omega_3$ we can easily construct $\Omega_6$:

$$\Omega_6 = \begin{pmatrix} \underline{0} & \underline{0} & \Omega_2 \\ \Omega_2 & \underline{0} & \underline{0} \\ \underline{0} & \Omega_2 & \underline{0} \end{pmatrix} \quad (13)$$

Here $\underline{0}$ is a zero matrix with dimension $2r$. Accordingly, the MDS embedding vectors become:

$$\beta_i^{6r} = (\beta_i^T, \mathbf{0}^T, \mathbf{0}^T, \mathbf{0}^T, \mathbf{0}^T, \mathbf{0}^T)^T \quad (14)$$

With this representation, the terms in the double $\sum$ in Eq.(10) are 0 when $u - v < 6$. So what happens when $-v \geq 6$, i.e. the two strings have more than 6 consecutive mismatches, if we apply $\Omega_6$ as the transformation matrix? It will introduce unwanted variance to the final distance only when $u - v = 6n$, where n is an integer. We name $u - v \neq 6n$ as the Non-Identity Condition (NIC). Depending on the vectors on each side of the matrix, the addition can be either

positive or negative. However, when comparing CDR3 sequences with more than 6 mismatches, it is usually not important what the exact distance between them is. This is because only the sequences with highest similarities will be selected as antigen-specific TCR clusters, and at the desirable cutoff of the alignment score, the number of mismatches between two CDR3 sequences is usually smaller than 3.

*Case 4*: Non-consecutive Mismatches.

Without losing generality, we assume that the two sequences of interest, $s_i$ and $s_j$, both with length $k$, differ at the first and last positions, such that: $s_i = A_{i_k}A_{i_{k-1}} \ldots A_{i_2}A_{i_1}, k \geq 3$ and $s_j = A_{j_k}A_{i_{k-1}} \ldots A_{i_2}A_{j_1}, k \geq 3$. The isometric coordinates after transformation are:

$$x_i = \sum_{l=1}^{k} \Omega^l \beta_{i_l}, l = 1, 2, \ldots, k$$

$$x_j = \sum_{l=2}^{k-1} \Omega^l \beta_{j_l} + \Omega^k \beta_{j_k} + \Omega \beta_{j_1}$$

Their distance can be calculated:

$$\|x_i - x_j\| = (\beta_{i_1} - \beta_{j_1})^T(\beta_{i_1} - \beta_{j_1}) + (\beta_{i_k} - \beta_{j_k})^T(\beta_{i_k} - \beta_{j_k}) - 2(\beta_{i_1} - \beta_{j_1})^T(\Omega)^T(\Omega^k)(\beta_{i_k} - \beta_{j_k}) \quad (15)$$

This is similar to *Case 3*, where we handle multiple consecutive mismatches, except that the number of cross terms is smaller, which can be written as:

$$-2(\beta_{i_1} - \beta_{j_1})^T(\Omega^{k-1})(\beta_{i_k} - \beta_{j_k})$$

When $\Omega$ is selected as $\Omega_6$, similar to *Case 3*, the cross term is always 0 as long as NIC is observed. However, if NIC is violated, i.e. the two mismatches are exactly six amino acids apart, the cross term becomes non-zero. We need to evaluate the impact of this term to the final outcome. First, if the cross term remains negative (with probability 1/2), the estimated isometric distance will be smaller than the exact value, which will not affect the outcome, as we will apply the stringent Smith–Waterman alignment to ensure high sequence similarity (section 3). It can be shown that, for CDR3s with length 16, with first 3 and last 2 clipped, the chance of having two mismatches exactly 6 amino acids apart is $\frac{5}{\binom{11}{2}} = 0.091$, which is the maximum among all the lengths. Therefore, violation of NIC will affect at most $0.091/2 = 4.6\%$ of the comparisons with two mismatches. When this happens, somewhat similar sequences will have larger distance and might be excluded from the downstream clustering. To mitigate this effect, we applied a relatively large default isometric distance cutoff (-t 10) to be inclusive. The current choice of parameterization is a balance between clustering accuracy and computation speed.

*Nearest neighbor centroid clustering.* The approximate isometric embedding of CDR3 sequences allows efficient search of their nearest neighbors (NN) in the Euclidean space for fast clustering. We used a python package Facebook AI Similarity Search, or faiss[46], to perform fast indexed-NN search. To find the nearest neighbor of one of the N numeric vectors in $\mathbb{R}^r$, the time complexity of *faiss* is $O(r\log(N))$.

We used the following strategy to divide the coordinates of CDR3s ($x$) into neighboring clusters. Before clustering, identical CDR3s were grouped together. First, for each unique sequence $x_i, i = 1, 2, \ldots, N$, we found its nearest neighbor $x_j, j = 1, 2, \ldots, N; j \neq i$. If the distance between $x_i$ and $x_j$ was within a user-defined cutoff (-t option, *thr*), the two points were merged as a new point, with the centroid $\frac{x_i + x_j}{2}$ as the new coordinate. If the distance exceeded the cutoff, both points were removed from iteration. There would be two types of removed points: (1) point containing only one CDR3 sequence; (2) point as a centroid of multiple CDR3s. A CDR3 pre-cluster was recorded for each of the second type of points. The above steps were repeated until the number of points reached to zero or did not further decrease. CDR3s with different lengths were separately clustered. All pre-clusters were kept for further filtering.

*K-mer guided fast Smith–Waterman alignment with TCR variable gene matching.* CDR3s from a pre-cluster are usually highly similar, but they may not qualify as antigen-specific groups because (1) sequences may not be similar enough due to imperfect isometric embedding; (2) TCR variable (TRBV) gene information was not taken into consideration. We therefore performed a filtering step to select antigen-specific CDR3 clusters based on Smith–Waterman alignment, and TRBV gene matching.

The size ($m$) of pre-clusters can be large, and direct pairwise comparison will result in quadratic complexity $O(m^2)$. We first applied the TRBV information to reduce cluster size. Specifically, we used a pre-calculated matrix[1] of alignment scores between a pair of TRBV alleles. For each pair of CDR3 sequences in the pre-cluster, we compared their TRBV alleles. If the score was above a user-defined (-G option, *thr_v*), an edge was added between the two sequences. We ran depth-first search (dfs) on the final graph to generate isolated subgraphs, with each subgraph a new pre-cluster. This step will split the original pre-cluster into several smaller ones.

Next, we used a k-mer approach to perform Smith–Waterman alignment. For each CDR3 sequence, we divided it into consecutive 5-mers. A k-mer dictionary was built to store all the sequences, with keys being unique 5-mers, and the values being the CDR3s that contain the given 5-mer. We allowed one mismatch in the

5-mer when building the dictionary. For example, sequence CASSGVTEAFF is indexed under both SSGVT and SSVAT. In this way, CDR3 sequences were connected into a graph via shared k-mers. For each edge in this graph, we ran Smith–Waterman alignment with BLOSUM62 substitution matrix and calculated the alignment score. If the score is below a user-defined cutoff (-S option, $thr\_s$), the edge will be removed. The actual complexity of this step can vary from $O(m)$ to $O(m^2)$. The worse scenario is reached when every pair of CDR3s in a pre-cluster share similar k-mer motifs. We then ran dfs on the final graph to generate the final CDR3 clusters and report them as the final output.

**Query of new TCR-seq samples against existing reference**. After the generation of TCR clusters of an input dataset, GIANA can perform query of additional TCRs to this data (the reference). In the query mode, GIANA requires the input of the query file(s), the original reference data, and clustered reference data. First, the original reference and query TCRs were converted into isometric coordinates. Fast nearest neighbor search by *faiss* was then implemented, but limited to the query TCRs. TCRs with distances smaller than a user-defined cutoff (-t option, $thr$) were exported into a separate file (tmp_query.txt). This file contains all the TCRs that could possibly cluster with the query sequences. GIANA clustering was subsequently performed on this file to generate the TCR clusters satisfying the stringent cutoffs for Smith–Waterman alignment. The query TCR clusters were then merged with the reference clusters in the following way: for each query cluster, if any of the sequence came from an existing cluster in the reference data, the two clusters will be merged. This step is to ensure the inclusion of all the neighboring TCRs in the reference data. Under two conditions when a query cluster did not contain any sequence in the reference cluster: (1) all TCRs in the query cluster were similar, but private to the query sample; (2) query TCR was similar to some very rare reference TCRs, which were not clustered with any other reference samples in the original clustering. Following either condition, the query cluster was included in the final output.

We evaluated the time cost of the query mode by generating reference data containing 200 K, 1 M, 2 M, 6 M, and 10 M TCRs. We scanned different sizes of the query data, including 10 K, 20 K, 30 K, 40 K, and 50 K TCRs. Each query file was clustered against each of the reference data using a MacBook Pro computer with 3.5 GHz Dual-Core Intel Core i7 processor, and 16GB 2133 MHz LPDDR3 memory. Elapsed time was estimated using the *time* module of python.

**GIANA with stacked-vector encoding (GIANAsv)**. After MDS embedding of the 20 amino acids, the easiest way to obtain an isometric representation of a CDR3 string: $s = A_1A_2 \ldots A_k, k \geq 5$, is to construct a "stacked vector", i.e. to concatenate the embedding vectors $\beta_i, i = 1, 2, \ldots, k$, in the same order. This approach is referred as GIANAsv. The stacked vector representation is $x = (\beta_1^T, \beta_2^T, \ldots, \beta_k^T)^T$. It is easy to prove that this representation satisfies all the desirable qualities of the three cases discussed in the above section. Theoretically, when focusing only on sequences with six or fewer mismatches, the two approaches are identical. When CDR3s have more than six mismatches, GIANAsv is more accurate, but this scenario is not relevant to this context. The disadvantage of GIANAsv is that the dimension $r_{GIANAsv}$ of the embedding vector is larger than that of GIANA ($r_{GIANA}$). For GIANA, $r_{GIANA} \equiv 6 \times 16 = 96$. For GIANAsv, $r_{GIANAsv}$ varies with different CDR3 length (typically 12–17 amino acids), which can be 2–3 times larger than $r_{GIANA}$. Increased dimensionality results in higher memory burden and longer computational time for *faiss*.

**Computational complexity comparison of GIANA, GIANAsv, iSMART, GLIPH2, and TCRdist**. Other published TCR clustering methods were downloaded to compare with GIANA, including iSMART, GLIPH2 and TCRdist. We excluded the original version of GLIPH[2] from this comparison, as GLIPH2 is an improved version with higher computational efficacy and covering all the functionalities of GLIPH. We used the TCR repertoire sequencing data of a healthy donor of a previous study[9], HIP13900, to test the performances. TCR clones were ordered based on their abundance, and the top 10 K, 20 K, …, 100 K sequences were selected. All five methods were applied to each of the subsamples. GIANA, GIANAsv, iSMART, and GLIPH2 were implemented using the default parameters. TCRdist does not provide clustering, and only pairwise distances were calculated under the default setting. All methods were run on a MacBook Pro computer with 3.5 GHz Dual-Core Intel Core i7 processor, and 16GB 2133 MHz LPDDR3 memory. All methods were applied using a single thread on CPU. When clustering the reference data of 10 million sequences, we implemented GIANA on the dataset using a high-performance computing (HPC) super cluster, with 128 G memory allocation and 8 CPU nodes. Links to the software compared in this work are listed below:

iSMART: https://github.com/s175573/DeepCAT/blob/master/iSMARTm.py
TCRdist: https://github.com/phbradley/tcr-dist
GLIPH2: http://50.255.35.37:8080/tools

**Estimation of clustering antigen-specificity of GIANA, iSMART, GLIPH2, and TCRdist**. TCRs specific to known epitopes were collected from the Immune Epitope Database and Analysis Resource (IEDB)[11], the VDJdb online browser[12], and recent literature[1,13]. We kept only TCRβ CDR3 sequences, TRBV genes, and their

associated antigens. After removal of redundant or incomplete sequences, we obtained a total of 61,366 CDR3s, covering ~900 epitopes from diverse pathogens. All four methods were applied to the dataset to perform antigen-specific clustering using their default parameters. For TCRdist, we wrote the R code to perform depth-first search on the sequence pairs with distances smaller than 11. To date, the time complexity calculation for TCRdist does not include the depth-first search to find TCR clusters. This cutoff of 11 has balanced sensitivity and specificity, comparable to that of iSMART. Choosing a larger cutoff will increase the total number of clustered TCRs, at the cost of lower specificity of each cluster.

In this work, we first defined the clusters with all the TCRs specific to the same antigen as "pure clusters". Pure cluster retention was defined as the total number of TCRs contained in all the pure clusters divided by the total number of sequences (60,700 after removal of epitopes with only 1 TCR):

$$\text{Retention} = \frac{\text{\#TCRs in all pure clusters}}{\text{\#Total TCRs}} \qquad (16)$$

We also defined pure cluster fraction as the number of pure clusters divided by the number of total clusters:

$$\text{Fraction} = \frac{\text{\#Pure clusters}}{\text{\#Total clusters}} \qquad (17)$$

Normalized mutual information (NMI) is defined as twice the mutual information between a TCR cluster and antigen specificity divided by the sum of their entropies. Some CDR3s may occur in multiple clusters in GLIPH2 output. We calculated the NMI for each TCR cluster and took the averaged value. Although clusters were not independent due to recurrent TCRs, this approach is asymptotically unbiased with bounded variance under the weak law of large numbers. The same approach was used for NMI calculation for all the methods.

**Performance evaluation of antigen-specific TCR identification in TCR-seq data**. We performed in silico mixing experiments to assess the performance of GIANA in finding TCRs specific to known antigens. We selected three antigens that are unlikely exposed to healthy donors: the YAW and YLQ epitopes from the Severe Acute Respiratory Syndrome Coronavirus-2 (SARS-CoV-2)[15] and the FRD epitope from the Human Immunodeficiency Virus-1 (HIV-1)[16]. TCRs specific to each epitope were selected, with redundancy removed. For each antigen, we randomly sampled 20% of TCRs (testing data) and mixed them with the 100 K sequences from healthy donor HIP13900. There was no overlap between the remaining 80% of antigen-specific TCRs (training data) and the testing data. The mixed sample was considered to be a pseudo-patient carrying the corresponding pathogen. We combined the mixed sample with the training data and applied GIANA with Smith–Waterman alignment score cutoff ($thr\_s$) ranging from 3.0 to 4.0 (0.1 increment). For each epitope and parameter setting, we ran 20 times of in silico mixing to capture the variations in the data.

From the resulting data, we evaluated the prediction performance. We selected the TCR clusters with at least one TCR from the training data. All the TCRs in these clusters, excluding training data, were positive calls. All TCRs that were not co-clustered with any training TCR were negative calls. True-positive calls were defined as sequences labeled as "testing data", whereas true-negative calls were sequences from the original 100 K TCRs of the healthy donor. Specificity was defined as the number of true-negative calls divided by 100 K. Sensitivity was defined as the number of true-positive calls divided by the total number of testing TCRs.

**Reference-based TCR repertoire classification**. To test the feasibility of repertoire classification using TCR clustering, we combined 10, 50, and 100 COVID-19 samples[20] (Table S4) with 10, 50, and 100 healthy controls[9] to generate 3 reference data with 20, 100, and 200 sample. Each sample contained 10 K TCRs, selected by ranking the clonal abundance. Query samples contained 154 COVID-19 patients[20] and 120 HCs[9]. There was no overlap between query and reference samples. We generated the TCR clusters for each query sample using GIANA. For each sample, we first removed TCR clusters with more than 100 samples, as these TCRs were likely generated from small-world connections and not informative to disease specificity[6]. For the remaining clusters, we calculated the fraction of reference TCRs contributed by the COVID-19 patients and used this quantity as the predictor.

In the multiple disease classification task, we first combined 712 cancer, 311 COVID-19, 25 multiple sclerosis (MS) patients, and 100 HC samples and produced a reference data of 10 M TCRs. We collected another 62 cancer, 193 COVID-19, 12 MS, and 153 HC samples to make the query, assuming the disease labels were unknown. Same analysis was performed for each query cluster file to estimate the fractions of each disease category, including HC. We used these fractions to predict diseases and performed the ROC analysis. Specifically, we used HC fractions for all the comparisons with HC samples. As an exploratory approach, for pairwise separation of the 3 diseases, we always used the difference between the two disease fractions. For example, when predicting cancer from MS patients, we used *Cancer Fraction-MS Fraction* as the predictor.

**Statistical analysis**. We used the pROC[47] package of the R programming language[48] to generate the ROC curves and estimate the AUC values, with 95% confidence intervals computed by 2,000 stratified bootstrap replicates, implemented using the ci.auc function in the pROC package. t-statistic of Fig. 3c was produced using the t.test function, to perform two sample t-test using the COVID-19 fractions to separate the COVID-19 and HC query samples. Normalized mutual information (NMI) of a method was calculated as the averaged NMI of all the clusters predicted by the method and the true antigen labels. NMI will be maximized if all TCRs in a cluster are specific to only one antigen. Figure 4a was generated using the igraph[49] package. Heatmap with annotated values was produced using heatmap.2 function in the gplots package. For all the boxplots displayed in the figures, the middle line defines the median value, with borders of the boxes indicating the 25% (Q1) and 75% (Q3) quartiles of the data. Lower and upper whiskers corresponded to Q1 – 1.5IQR and Q3 + 1.5IQR, where IQR is short for inter-quartile range.

**Software and package version information**. R and packages: R: 3.5.1, pROC: 1.13.0, igraph: 1.2.4.2, gplots: 3.0.1.1

Python and modules: Python: 3.7.3, numpy: 1.18.1, faiss: 1.5.1, pandas: 0.25.3, sklearn: 0.22.1, biopython: 1.76

**Reporting summary**. Further information on research design is available in the Nature Research Reporting Summary linked to this article.

## Data availability
The associated TCR-seq datasets, as well as metadata of the related samples are available at Zenodo[50]. Accession links of the original TCR-seq datasets are provided in Tables S4 and 5. Immune Epitope Database was accessed via: http://www.iedb.org/home_v3.php. VDJdb was accessed via: https://vdjdb.cdr3.net.

## Code availability
GIANA and GIANAsv source codes, R code to run GIANA visualization tree, and training datasets are available at: https://github.com/s175573/GIANA[51]

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

## Acknowledgements

This work is supported by the following funding sources: Cancer Prevention and Research Institute of Texas (CPRIT) RR170079 (B.L.), NCI 1R01CA245318 (B.L.), NIGMS 5R01GM126479 (X.Z.), and CPRIT RP190107 (X.Z.).

## Author contributions

B.L. and X.Z. conceived of this project. B.L. wrote the GIANA codebase. H.Z. and B.L. performed the benchmark analysis. B.L. wrote the manuscript together with X.Z. and H.Z.

## Competing interests

The authors declare no competing interests.
