## [Peer Review File · Nature Communications]

Reviewers' Comments:

Reviewer #1:

Remarks to the Author:

This study proposes a computationally efficient method GIANA to cluster large-scale TCR datasets. This is a timely study as immune repertoire data has been growing rapidly and there are increasing demands for efficiently analyzing such data. The main novelty of this paper is to map CDR3 sequences to a Euclidean space in which the distance between two sequences approximately reflects their amino acid dissimilarity. This mapping allows one to quickly identify and cluster similar sequences using a nearest neighbor search in the Euclidean space. The more time-consuming Smith-Waterman alignment can then be conducted only on pairs of similar sequences rather than all pairs of sequences. The authors performed benchmark analyses to evaluate computational efficiency (i.e., memory usage and runtime) and clustering performance (sensitivity and precision) of their method compared to three existing methods. They were able to demonstrate that (1) GIANA is scalable especially when applied to large-scale TCR sequencing data (10^5 to 1 million TCRs) and (2) can identify antigen-specific clusters with high specificity and precision. They also demonstrated the use of this method in querying TCR database and classifying patient samples based on the TCR repertoire.

Overall, the method presented here is novel and interesting. However, there are several issues that I suggest the authors to address or discuss in order to more convincingly establish the superiority of this new approach.

Major comments:

1. The isometric embedding in GIANA only considers distance calculation between two sequences of the same length. If two CDR3 sequences have different lengths (e.g., two sequences that are almost identical except that one sequence is 1 amino acid longer), it is unclear whether the GIANA embedding will appropriately characterize their similarity?
2. The GIANA method was described using consecutive mismatches as examples. It will be useful to also provide examples to explain how the method works when there are multiple mismatches that are not consecutive (e.g. CASSGVTEA vs. CASTGVAEA).
3. Since the numeric encoding scheme introduced in the Methods is the key for embedding CDR3 sequences, it will be helpful to use a few real amino acid sequences as examples to demonstrate the encoding process. This will help readers understand the procedure more easily.
4. When evaluating accuracy in predicting antigen-specific TCRs (Fig 1c-d), the authors used precision (= percentage of pure clusters in all clusters) and sensitivity (= total number of TCRs in all the pure clusters). These two criteria only provide a partial picture of the methods' performance. As a simple example, if all clusters are singletons (i.e., each cluster contains only one TCR, or equivalently a method does not do any clustering), then all clusters are pure, and therefore both the precision and sensitivity are perfect (= 1). However, this result is not necessarily what users want. Ideally, one would like to have TCRs with the same antigen-specificity to be grouped in only one cluster or in only a few clusters rather than in a large number of clusters, but this aspect of performance is not evaluated in this manuscript. It is therefore important to also evaluate different methods from the antigen perspective. For example, suppose for a given antigen there are N known antigen-specific TCRs. One natural question is how many pure clusters and pure TCRs are identified from these N TCRs? An ideal case would be: for each antigen the number of identified pure clusters is 1 or a small number but the clusters are still relatively pure. Based on Table S2, the number of pure clusters is quite large (>8000) and there are on average only 2 TCRs per pure cluster. With so many small pure clusters, it would be difficult for researchers to effectively explore the pure clusters in practice.
5. Fig1c-d compared GIANA with existing TCR sequence clustering methods through precision and sensitivity. However, specificity was not compared. What was the cutoff for Smith Waterman

alignment score used in Fig 1c-d? Since different cut-off values can produce different results, showing the performance as a curve (similar to ROC or precision-recall curves) would better illustrate the performance of different methods.

6. Fig 1e shows that different cutoffs of Smith-Waterman alignment score lead to different clustering performance. How should users select this cutoff when running GIANA on a new dataset? Similarly, how should users select the parameters in the nearest neighbor centroid clustering?

7. In the MDS approximation of BLOSUM62 matrix, please explain why the dimension of embedding space is chosen to be 16. Fig. S1 did not show performance of other dimensions, therefore it is unclear why 16 is the optimal dimension. Please provide details and supporting data for the claim that "Simulation suggests that the optimal dimension for isometric representation is 16 (Figure S1)".

8. Will GIANA's performance be affected by different choices of BLOSUM matrices? For example, what will happen if one switches to BLOSUM80 or BLOSUM50 matrix?

9. When comparing accuracy of predicting antigen-specific TCRs in Fig S5, GIANA is only compared with GLIPH2. What would be the performance of the other methods (iSMART and TCRdist)? Also, if possible, comparing the performance curves like ROC curves (by changing cutoffs of various parameters) would be more informative than comparing the results at one specific cutoff value.

10. All unique, non-redundant TCRs can be clustered into different groups based on similarity. Is there a way to visualize the output TCR clusters (i.e. network or tree) and quantify the relationship between clusters?

11. Please provide a list of antigen-specific TCRs along with its matched antigen pool in a Supplemental Table.

12. Please provide a detailed user manual for the GIANA software. For example, the parameters and its meanings.

Reviewer #2:
Remarks to the Author:
Summary

In this manuscript, Zhang and colleagues describe a method for very fast clustering of TCRs which is portrayed to be useful for antigen-specific TCR clustering as well as for multi-disease repertoire classification. The authors also provide a fast way for performing query-based database searches.

Overall opinion: Both fast clustering of TCRs as well as disease state classification are important problems that deserve investigation as provided in this manuscript. However, the authors fail to provide good argumentation of why their method is better than the state of the art both for the clustering and classification part of the paper. It further remains unclear how GIANA can improve our biological understanding of immune receptor biology.

Below, please find my comments per section.

Major

Results

- The terms precision and sensitivity have a standard meaning in classification settings, where they

are defined based on true/false positive/negatives. The current setting is not supervised, and the authors have chosen to still use these terms and define them in their own custom way. This can be misleading to readers. Indeed, it would seem that the current definitions tend to give higher scores for methods producing smaller clusters, though without this being entirely clear or well discussed. Since the sequences are annotated with specificity, it seems that the authors could indeed choose to define the problem as a supervised one, relying on the standard definitions of training and test. If the authors have good reasons to instead define it in the currently written manner, they should avoid using the standard terms that have existing connotations, and instead define their own custom terms to represent their custom measure of clustering quality. They should also discuss to what degree the results achieved for other methods are influenced by the current measure.

- When the problem is later in the manuscript viewed as a supervised learning problem, the accuracy of GIANA for various problems is not compared to any competing method - neither methods specifically designed repertoire classification nor any other clustering method used in a similar setup as GIANA.

- "isometric embedding of BLOSUM62 matrix" → can you discuss why BLOSUM62 and in what way is it crucial to the clustering?

- You state that GIANA "achieves higher accuracy in predicting antigen-specific TCRs" – in what way is this due to your specific algorithm? As I understand, it is completely unsupervised and thus your result is probably more due to the biological/technical structure of the data than to your algorithm? In other words, why do you expect all TCRs with a given epitope to cluster together? It has been shown that TCR recognizing the same epitope can be very different (Dash/Glanville, 2017, Nature) – is this also captured by GIANA? Have you investigated the misclassified ones to delineate the sequence features of those TCRs that do not co-cluster with their "epitope peers"? Can you explain why your method performs "better" than the other methods?

- With regard to tcrdist – did you benchmark against the newest iteration:

<https://github.com/kmayerb/tcrdist3>? Can you add all software versions you used to the Methods?

- Fig 1A: leaves completely opaque what the aim of your study is. Please devise a much clearer figure. Also, Fig S3 seems to be (nearly) identical to Fig 1A?

- Fig 1B: why do you put speed as 1B? Isn't it more intuitive to start with sens/spec of your method?

- Fig 1C-E please add schematics of how you measure precision/sensitivity/specificity as for now it remains completely opaque. And again, you are using an unsupervised method: how can you contribute any clustering performance to the method alone?

- Fig 2b: i think the usual use case is that the number of Ref TCRs is smaller (for example VDJdb) versus the number of query TCRs. can you show the graph when the numbers are exchanged between ref and query?

- Fig 2d: can you add to the Methods the adaptive study ids? Otherwise, it's completely unclear what data you have processed? Regarding the other data (cancer etc): have all studies been sequenced with the same adaptive protocol? Otherwise, they are not comparable. Please state the adaptive protocol in the methods also. Please also add the adaptive ids with links to the zenodo page you set up. What's the percentage of sequences that co-cluster with Covid-19 references? What order of magnitude difference to healthy are we talking about? Is this shown somewhere?

- Fig S8: you show Spearman correlation. What are you correlating?

- Fig 3: what if you choose something more difficult than leave one out? How does predictive performance change?

- Fig 4: "Since the query samples were derived from studies not included in the reference data, the high AUCs were not caused by unknown batch or cohort-specific effects, thus likely reflected the real predictability for the three diseases." This statement is very unclear to me. Studies sequenced with different protocols could very well explain high predictive performance.

- Please provide a reference for the following claim regarding repertoire classification: "The common limitation of these methods is their high computational cost that prevents from scaling up to large TCR-seq datasets"

- For the statement "Interestingly, lung cancer and COVID-19 patients formed a separate cluster", any result that is challenging to justify can be viewed as interesting, but it should also be discussed what this might be indicating in terms of weaknesses of the approach.

Introduction

- Please cite <https://pubmed.ncbi.nlm.nih.gov/30247624/> and discuss how your paper is different or similar in measuring cluster accuracy

Please cite: <https://www.biorxiv.org/content/10.1101/2021.02.22.432291v1> and discuss how your paper differs

- Please re-introduce the abbreviation of GIANA in the introduction and discuss to what extent the antigen-specific part of the name is relevant with respect to the clustering and prior literature

Discussion

- "In summary, GIANA is a novel antigen-specific TCR clustering algorithm that is able to efficiently handle tens of millions of sequences." Again, to what extent is GIANA built with clustering antigen specificity in mind? Specifically, what do we learn about antigen-specific patterns using GIANA?

- Can the authors discuss to what extent their method is dependent on the accuracy of the reference database as well as the metadata of the samples (age, sex, etc)? It is known these factors can influence the repertoire – was sex, age etc matched between HC, Covid and Cancer? If available can you provide the metadata in the supplementary? Similarly, was sequencing depth similar across the three immune states? To what extent is your method vulnerable to different size studies? Has the data of the studies in question been previously published? If so, cite them corresponding papers if not already done.

Minor

- Please add citations to graphics tools used for plotting to the Methods section

The authors use the pROC package but only cite R but not proc. [https://cran.r-](https://cran.r-project.org/web/packages/pROC/citation.html)

[project.org/web/packages/pROC/citation.html](https://cran.r-project.org/web/packages/pROC/citation.html). Please check if all other computational tools have been cited correctly.

Reviewer #3:

Remarks to the Author:

In this manuscript, the authors described a novel method GIANA to identify TCRs targeting the same antigens. GIANA implemented a smart algorithm to convert each CDR3 into a vector space so the sequence similarity can be approximated by Euclidean distance. The transformation allowed a highly efficient method to pre-cluster similar CDR3s and reduced the overall computational cost for TCR clustering and query. The authors not only showed improved accuracy and speed of GIANA against other state-of-the-art methods, but also demonstrated several biological applications such as identifying COVID-19 patients from blood TCRs. GIANA has been tested successfully on our system smoothly and produced meaningful results. However, there are a few technical issues that should be addressed or clarified before this paper can be published on Nature Communication.

1. The Euclidean distance between the transformed vectors for CDR3s is critical for clustering. What is the default distance threshold GIANA adopted to obtain the pre-clusters, and why is this value chosen? Even though the authors showed a strong correlation between the Euclidean distance of the transformed vectors and Smith-Waterman alignment scores, it would be more accurate to quantify the Euclidean distance approximation. For example, when fixing a CDR3 and a large set of CDR3s with desirable Smith-Waterman score, what is the proportion of those CDR3 with Euclidean distance higher than the default threshold (-t)?

Furthermore, the authors only mentioned that the Euclidean distance approximation is less accurate with more than six consecutive mismatches. However, the same issue happens if there are only two mismatches that are six amino acids apart, which is more common in practice. The experiment above

could also help explain such cases.

2. GLIPH2 may assign one CDR3 to multiple clusters, and some of the large clusters are pretty aggressive, which combines multiple more fine-grained clusters. Therefore, GLIPH2 has a low number of "pure clusters" and "pure TCR" when considering all clusters. Could the authors explain how to handle such cases when comparing the sensitivity and precision of GLIPH2 and GIANA?

3. The measure of precision and sensitivity using "pure cluster" may not be sufficient for the evaluations. For example, one method could call many small clusters, hence having a higher chance to obtain pure clusters and achieving higher performance. Measures like adjusted rand index or normalized mutual information could be more useful to compare the true TCR clusters and the predicted TCR clusters.

4. For the GIANA algorithm, the authors used G_2XG_3 to transform the amino acid representations on different positions to orthogonal spaces. Would the more naive transformation like shifting each vector to the next r position work?

5. In the abstract, the authors claimed GIANA could process 100 billion sequence comparisons within 3 minutes. However, this is misleading. With pre-cluster and k-mer guidance, GIANA conducted a much smaller number of sequence comparisons when querying 10^4 TCRs against 10^7 reference sequences. So the computational load is not equal to 100 billion pairwise comparisons.

Reviewer #4:

None

Summary of revision

In this revision, we addressed reviewers' suggestions by making the following major changes:

- 1) replace Figure 1a with a new schematic plot;
- 2) added per-epitope measures for all 4 methods;
- 3) added a section in Methods to describe the case with non-consecutive mismatches
- 4) added a "precision-recall" like curve to probe different -S options
- 5) compared original GIANA with the alternative using BLOSUM50
- 6) added a tree plot analysis to visualize output TCR clusters
- 7) compared GIANA query with DeepRC
- 8) compared GIANA with the latest version of TCRdist (tcrDist3)
- 9) applied cross-validation instead of leave-one-out to evaluate GIANA query on the training samples
- 10) added Normalized Mutual Information to compare the performances of the four methods.

Importantly, in the revised manuscript, we removed the language describing superior clustering accuracy of GIANA. Instead, our main message is that GIANA achieved higher scalability and computation speed. The changes in the manuscript has been highlighted with blue color. Below please find our point-to-point response to reviewers' comments, where the original comments are in blue color, and our responses in black color.

REVIEWER COMMENTS

Reviewer #1 (Expertise: Statistical algorithm development, Genomics data analysis, Immunogenomics):

This study proposes a computationally efficient method GIANA to cluster large-scale TCR datasets. This is a timely study as immune repertoire data has been growing rapidly and there are increasing demands for efficiently analyzing such data. The main novelty of this paper is to map CDR3 sequences to a Euclidean space in which the distance between two sequences approximately reflects their amino acid dissimilarity. This mapping allows one to quickly identify and cluster similar sequences using a nearest neighbor search in the Euclidean space. The more time-consuming Smith-Waterman alignment can then be conducted only on pairs of similar sequences rather than all pairs of sequences. The authors performed benchmark analyses to evaluate computational efficiency (i.e., memory usage and runtime) and clustering performance (sensitivity and precision) of their method compared to three existing methods. They were able to demonstrate that (1) GIANA is scalable especially when applied to large-scale TCR sequencing data (10^5 to 1 million TCRs) and (2) can identify antigen-specific clusters with high specificity and precision. They also demonstrated the use of this method in querying TCR database and classifying patient samples based on the TCR repertoire.

Overall, the method presented here is novel and interesting. However, there are several issues that I suggest the authors to address or discuss in order to more convincingly establish the superiority of this new approach.

We highly appreciate the reviewer's acknowledgement of the strength of GIANA and the accurate interpretations of our results. We have extensively revised the manuscript according to the reviewer's comments.

Major comments:

1. The isometric embedding in GIANA only considers distance calculation between two sequences of the same length. If two CDR3 sequences have different lengths (e.g., two sequences that are almost identical except that one sequence is 1 amino acid longer), it is unclear whether the GIANA embedding will appropriately characterize their similarity?

As the reviewer has correctly pointed out, currently GIANA only calculates distances between two CDR3s of the same length. In the scenario where two sequences differ by a gap, the isometric distance calculation will rely

on the position of the gap, and might be inaccurate. Therefore, in the current version of GIANA, the input sequences are split into smaller subsets by length, with each subset individually calculated for isometric distances and subsequently clustered. Consequently, we avoided direct comparison of two CDR3s of different lengths. This strategy does not affect the prediction accuracy as shown in our benchmark. With the stringent Smith-Waterman score cutoff for antigen-specific clustering, majority of the clusters are of the same length, as gap penalty in the protein substitution matrix is usually larger than any mismatches. This is also consistent with the observation that although TCRdist considers CDR3s of different length, 99% of its output are clusters of the same length. This point is discussed in our revised manuscript.

2. The GIANA method was described using consecutive mismatches as examples. It will be useful to also provide examples to explain how the method works when there are multiple mismatches that are not consecutive (e.g. CASSGVTEA vs. CASTGVAEA).

We added a section in the Methods to extensively discuss the mathematical basis when calculating the isometric distance with non-consecutive mismatches:

“Without losing generality, we assume that the two sequences of interest, s_i and s_j , both with length k , differ at the first and last positions, such that: $s_i = A_{i_k}A_{i_{k-1}} \dots A_{i_2}A_{i_1}$, $k \geq 3$ and $s_j = A_{j_k}A_{j_{k-1}} \dots A_{j_2}A_{j_1}$, $k \geq 3$. The isometric coordinates after transformation are:

$$\begin{aligned} x_i &= \sum_{l=1}^k \Omega^l \beta_{i_l}, l = 1, 2, \dots, k \\ x_j &= \sum_{l=2}^{k-1} \Omega^l \beta_{j_l} + \Omega^k \beta_{j_k} + \Omega \beta_{j_1} \end{aligned}$$

Their distance can be calculated:

$$\|x_i - x_j\| = (\beta_{i_1} - \beta_{j_1})^T (\beta_{i_1} - \beta_{j_1}) + (\beta_{i_k} - \beta_{j_k})^T (\beta_{i_k} - \beta_{j_k}) - 2(\beta_{i_1} - \beta_{j_1})^T (\Omega)^T (\Omega^k) (\beta_{i_k} - \beta_{j_k})$$

This is similar to *Case 3*, where we handle multiple consecutive mismatches, except that the number of cross terms is smaller, which is:

$$-2(\beta_{i_1} - \beta_{j_1})^T (\Omega^{k-1}) (\beta_{i_k} - \beta_{j_k})$$

When Ω is selected as Ω_6 , similar to *Case 3*, the cross term is always 0 as long as Non-Identity Condition, or NIC ($k - 1 \neq 6n$, n is an integer) is satisfied. However, if NIC is violated, i.e. the two mismatches are exactly 6 amino acids apart, the cross term becomes non-zero. We need to evaluate the impact of this term to the final outcome. First, if the cross term remain negative (with probability 1/2), the estimated isometric distance will be smaller than the exact value, which will not affect the outcome, as we will apply the stringent Smith-Waterman alignment to ensure high sequence similarity (section 3). It can be shown that, for CDR3s with length 16, with first 3 and last 2 clipped, the chance of having two mismatches exactly 6 amino acids apart given there are two mismatches is $\frac{5}{\binom{11}{2}} = 0.091$, which is the maximum

probability among all the lengths. Therefore, violation of NIC will affect at most $0.091/2 = 4.6\%$ of the comparisons with two mismatches. When this happens, somewhat similar sequences will have larger distance and might be excluded from the downstream clustering. To mitigate this effect, we applied a relatively large default isometric distance cutoff (-t 10) to be inclusive. The current choice of parameterization is a balance between clustering accuracy and computation speed.”

In fact, with -t = 10, we found that all sequence pairs with high SW alignment scores are included in the downstream clustering analysis. Please also see our response to Reviewer #3’s first comment for details.

3. Since the numeric encoding scheme introduced in the Methods is the key for embedding CDR3 sequences, it will be helpful to use a few real amino acid sequences as examples to demonstrate the encoding process. This will help readers understand the procedure more easily.

Following this comment, we made an example to demonstrate the encoding process as a Jupyter notebook. Using the mathematical framework described in the manuscript, we provided the python code to obtain the numeric coordinates of an input CDR3 step by step. This illustration is available in the GIANA website: https://github.com/s175573/GIANA/blob/master/example_of_CDR3_encoding.ipynb

4. When evaluating accuracy in predicting antigen-specific TCRs (Fig 1c-d), the authors used precision (= percentage of pure clusters in all clusters) and sensitivity (= total number of TCRs in all the pure clusters). These two criteria only provide a partial picture of the methods' performance. As a simple example, if all clusters are singletons (i.e., each cluster contains only one TCR, or equivalently a method does not do any clustering), then all clusters are pure, and therefore both the precision and sensitivity are perfect (= 1). However, this result is not necessarily what users want. Ideally, one would like to have TCRs with the same antigen-specificity to be grouped in only one cluster or in only a few clusters rather than in a large number of clusters, but this aspect of performance is not evaluated in this manuscript. It is therefore important to also evaluate different methods from the antigen perspective. For example, suppose for a given antigen there are N known antigen-specific TCRs. One natural question is how many pure clusters and pure TCRs are identified from these N TCRs? An ideal case would be: for each antigen the number of identified pure clusters is 1 or a small number but the clusters are still relatively pure. Based on Table S2, the number of pure clusters is quite large (>8000) and there are on average only 2 TCRs per pure cluster. With so many small pure clusters, it would be difficult for researchers to effectively explore the pure clusters in practice.

First, to clarify, the GIANA output does not contain any singletons. In our calculations of sensitivity and precision, each TCR cluster contains at least 2 CDR3 sequences. Therefore, the performance comparison of GIANA is based on informative TCR clustering results only.

We followed reviewer's suggestion to evaluate different methods from the antigen perspective. For each epitope, let N denote the number of antigen-specific TCRs. We calculated the following quantities to describe the distribution of pure clusters and pure TCRs: 1) number of pure clusters divided by N/1000 (Pure Cluster Count per Kilo TCRs), 2) number of pure TCRs divided by N/1000 and 3) averaged pure cluster size. As shown in the figure below, for both pure cluster count and pure TCR count, GIANA is highly correlated to TCRdist and iSMART, suggesting that the performance of antigen-specific calling of the three methods are similar. GLIPH2 and GIANA are also similar, while differed for a subset of antigens. This is mainly because motif-bases search (GLIPH2) can be sensitive to TCRs specific to a few epitopes (y-axis), but less specific to most of the other antigens (x-axis).

Figure for review 1: Various measures developed to compare different methods from the antigen perspective. On the scatter plot, each antigen is represented as a point. For a given antigen with N TCRs, “Pure cluster per kilo reads” is the number of 100% clusters divided by N/1000. “Pure TCR count per kilo reads” is the number of TCRs pooled from all pure clusters divided by N/1000.

For all four methods, the mean sizes of pure clusters are close to 2, indicating that the majority of the clusters contained only 2 TCRs. We agree with the reviewer that in the ideal scenario, all the TCRs specific to one epitope should be clustered into only a few medium to large sized clusters. However, previous studies have shown that TCRs specific to the same antigen can be very different (Dash et al., 2017 Nature; Glanville et al., 2017 Nature). Such dissimilarities represented distinct docking strategies of T cell receptors binding to the same epitope. For example, TCRs specific to the influenza M1 epitope (GIL) usually contain the classic RSS/RSA motif in the CDR3 loop, but a related study (Yang et al., 2017 J. Bio Chem) revealed that the LGGW motif can also bind strongly from a different direction. Such structural diversity cannot be captured by simple Smith-Waterman alignment. Consequently, CDR3s with dissimilar motifs will be fragmented into smaller clusters despite their shared specificity. We acknowledge this fact as a major limitation to all the concurrent computational methods, which necessitates future methodology improvements in this field. The above arguments have been added to the Discussion section of the revised manuscript.

5. Fig1c-d compared GIANA with existing TCR sequence clustering methods through precision and sensitivity. However, specificity was not compared. What was the cutoff for Smith Waterman alignment score used in Fig 1c-d? Since different cut-off values can produce different results, showing the performance as a curve (similar to ROC or precision-recall curves) would better illustrate the performance of different methods.

It is not feasible to define specificity in the benchmark dataset, because all the TCRs are specific to known epitopes. Consequently, there is no False Positive calls to substantiate specificity calculation. We apologize for this lack of clarity, and have revised the manuscript to avoid confusions. By default, the cutoff for SW alignment in GIANA was 3.5 (default parameter values are available at <https://github.com/s175573/GIANA>).

We thank the reviewer for the great suggestion to scan a range of parameters and obtain precision-recall curves, which leads to a more fair comparison between the methods and a performance boost of GIANA. This analysis was applicable to GIANA, TCRdist and iSMART, as all three are based on SW alignment. By changing the cutoff values of the alignment score, we obtained the following curves (Figure S5):

Figure for review 2: Precision-Recall curves comparing the performance of GIANA, iSMART and TCRdist.

At precision > 0.95 , all three methods share similar curves. The ‘elbow’ shape of the curves is due to the use of pure cluster TCRs in the calculation of recall. When the cutoff is reduced, TCRs from different antigens may get clustered, thus reducing the fraction of pure clusters. Previously we used a cutoff of 3.5 for both GIANA and iSMART. Based on this analysis, a cutoff of 3.6 is better, as it only slightly reduced the recall (from 0.268 to 0.267), but increased almost 3 percent of precision (from 0.932 to 0.961). Therefore, we changed the default parameter for -S option in GIANA to 3.6. The cutoff of TCRdist was chosen to be 15, but according to this analysis, we revised it to be 11, which yielded a higher precision (0.974) and similar level of recall (0.254). After updating the results related to Table S2 accordingly, we also revised our conclusion into “*GIANA achieved higher computational efficiency and scalability while maintaining the same level of clustering antigen-specificity*”.

6. Fig 1e shows that different cutoffs of Smith-Waterman alignment score lead to different clustering performance. How should users select this cutoff when running GIANA on a new dataset? Similarly, how should users select the parameters in the nearest neighbor centroid clustering?

With the calculation of the precision-recall curves for all 3 SW alignment based algorithms, we have adjusted the default cutoff to be 3.6, instead of 3.5. This cutoff offers a high precision (~ 0.96) and a relatively high sensitivity (~ 0.27) based on the averaged estimations by pooling 295 different antigens in the benchmark dataset. When running a new dataset, the users can reference the precision-recall curve of GIANA to choose the appropriate cutoff depending on the research purposes. For example, when performing repertoire classification tasks for complex diseases, such as cancer, where multiple disease-associated antigens are anticipated, it is usually more desirable to choose a lenient cutoff to include as many antigen-specific TCRs as possible. When dealing with diseases bearing immunodominant epitopes (such as influenza), users can usually afford to choose a more stringent cutoff to ensure higher specificity.

The cutoff for isometric distance (-t option), on the other hand, is less important for the outcome of GIANA. By default, we chose a distance cutoff (10) that is large enough to include more dissimilar CDR3s, which will be curated with SW alignment. With a more stringent cutoff, GIANA will become faster, as it processes fewer TCRs. Consequently, it has lower sensitivity even with a more lenient cutoff (-S option) for the SW alignment score. In sum, the -t option marks a hard boundary on the TCRs to be passed forward for clustering, where the -S option further refines these TCRs. It is usually not necessary to reduce the default value of isometric distance

cutoff unless there is an obvious need for higher (2X) speed. We have added the above guidance to the user manual of GIANA, in section “Choosing desirable parameters for TCR clustering”.

7. In the MDS approximation of BLOSUM62 matrix, please explain why the dimension of embedding space is chosen to be 16. Fig. S1 did not show performance of other dimensions, therefore it is unclear why 16 is the optimal dimension. Please provide details and supporting data for the claim that “Simulation suggests that the optimal dimension for isometric representation is 16 (Figure S1)”.

After transformation of BLOSUM62 into a pseudo-distance matrix as described in the Methods, the new matrix is not fully ranked. With classic MDS, the maximum dimension for the embedding space is 13. We applied the non-metric MDS calculation in python, and could explore dimensionality higher than 13. To maximize the embedding isometry, we first selected 2,300 training TCRs of length 14 from a previously described TCGA dataset (Zhang et al., 2020, Clin Can Res), and calculated the pairwise SW alignment scores. We applied the MDS to obtain isometric embedding vectors of different dimensions, ranging from 13 to 19. For each length, we calculated the Euclidean coordinates of the CDR3 sequences as described in the GIANA method, and compared the pairwise distance with SW scores. The maximum score was observed with dimension 16 (Spearman’s $\rho = -0.973$, Figure S2). We have added the above descriptions in the Methods section of the revised manuscript.

8. Will GIANA’s performance be affected by different choices of BLOSUM matrices? For example, what will happen if one switches to BLOSUM80 or BLOSUM50 matrix?

We followed reviewer’s suggestion by applying BLOSUM50 instead of BLOSUM62, and scanned a range of cutoffs for SW alignment to obtain the precision-recall curve. The new results were compared to the previous version of GIANA with BLOSUM62, as well as TCRdist and iSMART (figure below).

Figure for review 3: Precision-Recall curves comparing the performance GIANA (with BLOSUM62), iSMART, TCRdist and GIANA (with BLOSUM50).

“GIANA 50” is the curve for GIANA with BLOSUM50 matrix. The curve is very similar to the original version of GIANA with BLOSUM62 matrix, which is not surprising, because BLOSUM50 and BLOSUM62 matrices are similar in their off-diagonal values. The differences in the diagonal values were eliminated when transformed into distance matrix in GIANA. Based on this analysis, we conclude that the clustering accuracy of GIANA is relatively robust to the choice of protein substitution criteria. Therefore, the choice of either BLOSUM62 or BLOSUM50 would not affect the precision or recall of the final output.

9. When comparing accuracy of predicting antigen-specific TCRs in Fig S5, GIANA is only compared with GLIPH2. What would be the performance of the other methods (iSMART and TCRdist)? Also, if possible, comparing the performance curves like ROC curves (by changing cutoffs of various parameters) would be more

informative than comparing the results at one specific cutoff value.

The task in Fig 1e and Fig S5 involves handling of millions of TCRs. A total of 881 runs on 100K sequences would be needed. Currently, only GIANA and GLIPH2 can process such large scale dataset with affordable time. For example, it will take TCRdist over 100 days to finish this task, which is not practical. In addition, the TCR repertoire samples in the real world can be significantly larger than 100K. Therefore, we did not include iSMART or TCRdist in the comparison.

We agree with the reviewer that ROC curves would be more informative to compare different methods. However, to our best knowledge, GLIPH2 does not provide a tunable parameter that is similar to the SW alignment score, which can be changed to produce an ROC curve. In fact, we could not find any descriptions of the parameters used in the GLIPH2 website either. There is an interactive website where users can submit data to obtain GLIPH clusters (<http://50.255.35.37:8080/create>), but no parameter except for input data information was adjustable. Therefore, it is currently not feasible for us to create an ROC curve for GLIPH2.

10. All unique, non-redundant TCRs can be clustered into different groups based on similarity. Is there a way to visualize the output TCR clusters (i.e. network or tree) and quantify the relationship between clusters?

We followed reviewer's suggestion by adding -M option to GIANA to output the isometric coordinates of each CDR3 sequence. The coordinate matrix is available for visualizing the TCR clusters when combined with the standard GIANA output. One way to quantify the relationship between clusters is neighbor joining tree, using the pairwise distance of TCR sequences. Below is an example, showing the hierarchical relationships between clusters of TCRs specific to the influenza M1 epitope GILGFVFTL:

Figure for review 4: Tree plot of selected TCR clusters from GIANA output. Neighbor joining tree was calculated based on the isometric coordinates as part of the GIANA output. Each color indicates a TCR cluster. All TCRs were specific to influenza M1 GIL epitope.

For users to perform the above analysis, we provided a fully-documented R markdown file and uploaded to the github (<https://htmlpreview.github.io/?https://github.com/s175573/GIANA/blob/master/GIANAtree.html>).

11. Please provide a list of antigen-specific TCRs along with its matched antigen pool in a Supplemental Table.

Due to its size, we provided the list of antigen-specific TCRs along with the match antigens as a Supplementary Dataset and uploaded to the GIANA website:

https://github.com/s175573/GIANA/blob/master/data/TCRantigenData_unique.txt

12. Please provide a detailed user manual for the GIANA software. For example, the parameters and its meanings.

We have updated the GIANA github webpage (<https://github.com/s175573/GIANA>) to incorporate the descriptions of all the parameters.

Reviewer #2 (Expertise: TCR-antigen identification):

Summary

In this manuscript, Zhang and colleagues describe a method for very fast clustering of TCRs which is portrayed to be useful for antigen-specific TCR clustering as well as for multi-disease repertoire classification. The authors also provide a fast way for performing query-based database searches.

Overall opinion: Both fast clustering of TCRs as well as disease state classification are important problems that deserve investigation as provided in this manuscript. However, the authors fail to provide good argumentation of why their method is better than the state of the art both for the clustering and classification part of the paper. It further remains unclear how GIANA can improve our biological understanding of immune receptor biology.

We thank the reviewer for taking efforts to evaluate our work, and have extensively revised the manuscript to lift potential confusions and clearly deliver the conclusions from our results.

Below, please find my comments per section.

Major

Results

- The terms precision and sensitivity have a standard meaning in classification settings, where they are defined based on true/false positive/negatives. The current setting is not supervised, and the authors have chosen to still use these terms and define them in their own custom way. This can be misleading to readers. Indeed, it would seem that the current definitions tend to give higher scores for methods producing smaller clusters, though without this being entirely clear or well discussed. Since the sequences are annotated with specificity, it seems that the authors could indeed choose to define the problem as a supervised one, relying on the standard definitions of training and test. If the authors have good reasons to instead define it in the currently written manner, they should avoid using the standard terms that have existing connotations, and instead define their own custom terms to represent their custom measure of clustering quality. They should also discuss to what degree the results achieved for other methods are influenced by the current measure.

We agree with the reviewer that measure of clustering quality is not trivial, and it is necessary to clearly define the terminologies in the manuscript to avoid potential confusions. In the revision, we redefined the terms precision and sensitivity following a previous work (Meysman et al., 2019): 1) Pure Cluster Fraction (instead of clustering precision): the percentage of 100% pure clusters among all the TCR clusters returned by GIANA; 2) Pure Cluster Retention (instead of sensitivity): the total amount of TCRs in all the 100% pure clusters divided by the size of the entire dataset. To date, these changes do not affect any of our results, or conclusions in the downstream analysis.

We continued to use these measures because we want to measure the TCR clusters that could be unambiguously assigned to an antigen, which is one of the most important purpose to perform the clustering analysis on TCR repertoire samples. Higher fraction of “pure” TCRs in the output implicates a lower noise when using the TCR clusters to perform tasks that require higher specificity, such as TCR retrieval for single epitope (Figure 1e, now Fig 2d) or repertoire classification (Figure 3-4, now Fig 4-5).

That being said, we agree with the reviewer that “pure cluster” usually requires stringent cutoff of the Smith-Waterman alignment scores, which would result in more fragmented TCR clusters. In our comparison, TCRdist, iSMART and GIANA all rely on SW alignment, and thus favoring smaller clusters when higher specificity is desirable. On the other hand, GLIPH2 relies on the less-stringent motif-based search, which uses partial CDR3 sequences. Therefore, it creates larger TCR clusters with lower purity. Consequently, when choosing the pure clusters to evaluate clustering quality, GLIPH2 performs worse than the other 3 methods. Considering this potential bias against GLIPH2, we added a quality measure using the normalized mutual information (Fig 2c), yet still observed lowest value for GLIPH2. Despite these results, it is shown that GLIPH2 could be more inclusive, and more powerful for other tasks, such as comprehensively searching for TCRs related to complex pathogens, i.e. Mycobacterium tuberculosis (Huang et al., 2020, Nat Biotech). We have revised the Discussion to include these points and lift potential confusions.

- When the problem is later in the manuscript viewed as a supervised learning problem, the accuracy of GIANA for various problems is not compared to any competing method - neither methods specifically designed repertoire classification nor any other clustering method used in a similar setup as GIANA.

We understand that the reviewer would like us to perform repertoire classification using other clustering method (tcrdist/iSMART/GLIPH2), or specifically designed methods. First, in order to implement repertoire referencing, we developed a new functional module to perform query of new TCR repertoire samples, which is a non-trivial task. In addition, even if one developed such a module for iSMART or TCRdist, they were not suitable for repertoire query due to low computational efficiency. For GLIPH2, there is no existing options in the method to compare different repertoires either. Therefore, other clustering methods are not ready for this task in a similar setup as GIANA.

That being said, we agree with the reviewer that the accuracy of GIANA could be compared to tools specifically for repertoire classification. Currently, only two methods are designed for this task to our best knowledge: 1) Ostmeyer et al., 2019; 2) DeepRC (Widrich et al., 2020). To date, DeepRC has not been peer reviewed, and we are comparing to the preprint version on bioRxiv (<https://www.biorxiv.org/content/10.1101/2020.04.12.038158v3>). Both methods were based on multiple instance learning (MIL) and requires fitting of cohort-specific parameters. While our approach does not require any parameter fitting, we need to find a suitable reference data that has the same attributes as the query samples. For example, when predicting COVID-19 patients, our reference contains repertoire samples from true COVID-19 patients and negative controls. It would usually take efforts to set up and train different MIL algorithms. Fortunately, in Widrich paper, the authors performed comprehensive comparisons with Ostmeyer’s approach, and several variations of the deep neural network, using the Emerson et al., 2017 cohort containing both HCMV+ and HCMV- subjects as benchmark. We applied the same dataset, by applying 75% of the samples as the reference (similar as training), with the remaining 25% as test data. Each test sample was queried against the reference data. For each query sample, we calculated the fraction TCRs co-clustered with HCMV+ reference subjects, and used this fraction as a predictor. This simple approach reached 83.06% AUC (figure below), the same as the best strategy reported in the Widrich paper (table below). Therefore, we believe GIANA query is a competitive method for repertoire classification.

Method	AUC
GIANA	0.831
DeepRC	0.831
SVM (MM)	0.825
SVM (J)	0.546
KNN (MM)	0.679
KNN (J)	0.534
MIL (KMER)	0.582
MIL (TCRB)	0.515

Figure for review 5: Performance comparison between GIANA query and other methods reported in the DeepRC manuscript. Left: ROC curve of GIANA query. Right: AUC values for diverse methods. All AUCs were evaluated using the Emerson et al 2017 dataset, predicting the HCMV serology status. MIL (KMER) and MIL (TCRB) refer to the method developed in Ostmeyer et al., 2019.

- “isometric embedding of BLOSUM62 matrix” → can you discuss why BLOSUM62 and in what way is it crucial to the clustering?

The BLOSUM62 protein substitution matrix has been proven useful in numerous studies on protein sequences. It reliably reconstitute the functional similarities between different residues, which is desirable to our purpose. In the past, it has been routinely used in computational methods analyzing the TCR/pMHC complex. For example, the heavily-cited NetMHC tools developed by Nielsen group implemented a BLOSUM62-based artificial neural network model, which remains one of the best algorithm in this field. Furthermore, two previous work, TCRdist and iSMART also used BLOSUM62, and generated meaningful results of antigen-specific TCR clusters (Dash et al., 2017; Zhang et al., 2020). As an alternative, we also explored BLOSUM50 as substitution matrix, and observed similar clustering accuracy (please see our response to Reviewer #1’s 8 comment). Thus, use of different BLOSUM matrices is likely to generate very similar clustering results. Due to these reasons, we have continued to use BLOSUM62 in GIANA.

- You state that GIANA “achieves higher accuracy in predicting antigen-specific TCRs” – in what way is this due to your specific algorithm? As I understand, it is completely unsupervised and thus your result is probably more due to the biological/technical structure of the data than to your algorithm? In other words, why do you expect all TCRs with a given epitope to cluster together? It has been shown that TCR recognizing the same epitope can be very different (Dash/Glanville, 2017, Nature) – is this also captured by GIANA? Have you investigated the misclassified ones to delineate the sequence features of those TCRs that do not co-cluster with their “epitope peers”? Can you explain why your method performs “better” than the other methods?

The reviewer is correct in that TCR clustering based on sequence similarity does not take into account any antigen-specific information. Given the fact that dissimilar TCRs can recognize the same antigen, there is no guarantee that clustering would lead to higher antigen-specificity. In fact, after re-evaluating different methods by changing the cutoff of Smith-Waterman alignment score, we obtained the full spectrum of Pure Cluster Fraction and Retention (Figure S5). This analysis revealed similar clustering accuracy of GIANA compared to TCRdist or iSMART at stringent SW score cutoffs. We also investigated the sequences that do not co-cluster with other TCRs, and as expected, most of them contain distinct sequence motifs. In the revision, we carefully removed the claim that GIANA performs better than the other methods, and replaced with “achieves the same level of accuracy” when comparing the prediction performance.

- With regard to tcrdist – did you benchmark against the newest iteration: <https://github.com/kmayerb/tcrdist3>? Can you add all software versions you used to the Methods?

We used the original TCRdist with modifications to reduce memory consumption. The newest version provides a formal wrapper to the same core functions of the original TCRdist, and applies Numba to speedup. Consequently, the distance measures of TCRdist3 are identical to the original version, but the computational efficacy is improved. We implemented TCRdist3 as instructed (<https://tcrdist3.readthedocs.io/en/latest/tcrdistances.html>) to measure its speed in our benchmark dataset with 10K, 20K, ..., 100K, on the same computer where other methods were evaluated. We observed significantly higher memory usage of TCRdist3, due to its attempt to save the entire pairwise distance matrix, with $O(N^2)$ complexity. At default parameterization, TCRdist3 quickly consumes over 40G memory when analyzing only 20K sequences, and became impossible to run with more than 40K TCRs. At this setting, TCRdist3 is marginally faster (1.1X speedup) than TCRdist (table below).

Methods	10K	20K	30K	40K
---------	-----	-----	-----	-----

GIANA	1.5s	2.9s	4.2s	6.3s
TCRdist	145.9s	580s	1300s	2330s
TCRdist3	35.9s	313s	1110s	2080s

Figure for review 6. Comparison of computational efficiency of GIANA, TCRdist and the newest iteration TCRdist3. TCRdist3 fails to run on the same computer where other methods were evaluated with over 40K reads due to excessive memory consumption.

We believe that TCRdist3 is not suitable for the task outlined in this manuscript, because it is not feasible to use TCRdist3 to analyze ultra-large-scale TCR repertoire datasets (with over 10^6 sequences) due to excessive memory consumption. In this regard, the modified version of TCRdist tested in our benchmark is much more portable, with approximately the same time complexity.

Following the reviewer’s suggestions, we have added all the software links in the Methods section.

- Fig 1A: leaves completely opaque what the aim of your study is. Please devise a much clearer figure. Also, Fig S3 seems to be (nearly) identical to Fig 1A?

We followed reviewer’s comment and designed a clearer figure to illustrate the workflow of GIANA and made it Figure 1 as a separate figure to deliver the aim of our study:

Figure for review 7: Updated schematic plot of GIANA.

Figure S3 is an illustration of GIANAsv, which is another realization of isometric encoding. Instead of using the cyclic group encoding, we concatenated the MDS vectors of the amino acids in the CDR3 sequence, to construct a high-dimensional representation. GIANAsv is described in detail in the Methods and compared with other approaches in this work as well.

- Fig 1B: why do put speed as 1B? Isn’t it more intuitive to start with sens/spec of your method?

The primary goal of this work is to provide a fast TCR clustering method that scales up to 10^7 sequences, while maintaining the same level of clustering accuracy as the existing approaches. Therefore, computational efficiency is the most important aspect of GIANA, which was displayed as the first result in the manuscript.

- Fig 1C-E please add schematics of how you measure precision/sensitivity/specificity as for now it remains completely opaque. And again, you are using an unsupervised method: how can you contribute any clustering performance to the method alone?

As described in the response to the reviewer’s first comment, we have removed the terms of “precision” and “sensitivity” in Fig 1c (now Fig 2b) to avoid any confusions. We added equations in the Methods section to describe the definitions of pure cluster fraction and retention:

“In this work, we first defined the clusters with all the TCRs specific to the same antigen as “pure clusters”. Pure cluster retention was defined as the total number of TCRs contained in all the pure clusters divided by the total number of sequences:

$$Retention = \frac{\# TCRs \text{ in all pure clusters}}{\# Total TCRs}$$

We also defined pure cluster fraction as the number of pure clusters divided by the number of total clusters:

$$Fraction = \frac{\# Pure clusters}{\# Total clusters}$$

”

For Fig 1e (now Figure 2d), the setting is actually a supervised task. Here we defined the true positives as the TCRs with known specificity to YAW, YLQ or FRD. We spiked in a fraction of these TCRs in a real TCR repertoire, and used the remaining non-overlapping sequences to retrieve these TCRs. All non-spike-in sequences in that repertoire are considered true negatives, as the sample was HIV negative and was collected prior to COVID-19 pandemic. Therefore, we believe that the classic definition of sensitivity and specificity can reflect the accuracy of GIANA for this task.

- Fig 2b: i think the usual use case is that the number of Ref TCRs is smaller (for example VDJdb) versus the number of query TCRs. can you show the graph when the numbers are exchanged between ref and query?

We agree with the reviewer that our analysis is not the “usual” way for TCR repertoire classification. In the example of VDJdb, the interest is to learn epitope specificity through clustering the TCRs in an unknown repertoire. This application is discussed in Fig 1e (now Fig 2d). However, our goal is to assign disease labels to each repertoire sample by comparing query samples with existing reference through co-clustering. Therefore, each TCR cluster should be considered as a predictive feature in supervised machine learning. Although no parameter or model needs to be fit, the reference here has similar utility as training data, where features are learned from, and query sample labels are the predictions. Following the rationale of machine learning strategies, to avoid overfitting and to maximize predictability, it is always desirable to have more training data than testing data, and that is why we tested the scenario where reference number is greater than query. Nonetheless, we followed the reviewer’s suggestion and investigated the opposite scenario, by using the antigen-specific TCRs as reference (n=60K), and tested query samples with 60K, 70K, 80K, 90K and 100K TCRs. The results are similar to the scenarios tested in our manuscript. The graph is shown as below:

Figure for review 8: Time complexity of GIANA query mode in the scenario with more query TCRs than reference data.

- Fig 2d: can you add to the Methods the adaptive study ids? Otherwise, it’s completely unclear what data you have processed? Regarding the other data (cancer etc): have all studies been sequenced with the same adaptive protocol? Otherwise, they are not comparable. Please state the adaptive protocol in the methods also. Please also add the adaptive ids with links to the zenodo page you set up. What’s the percentage of sequences that co-

cluster with Covid-19 references? What order of magnitude difference to healthy are we talking about? Is this shown somewhere?

We added the download links in the immuneACCESS database to all the datasets used in this study (Table S4-5). All the cohorts in our study were generated using the immunoSEQ platform, processed using the Human T-Cell Receptor Beta (hsTCRB) kit. We added this information to Methods to lift potential confusions. We also added the download links to Zenodo webpage (doi: 10.5281/zenodo.4698929).

There are 40 – 45% of TCRs that co-clustered with Covid-19 reference samples for query Covid-19 patients, where for the query healthy controls, which is 4 – 9% lower (figure below, also Figure S7a):

Figure for review 9: Boxplot showing the distribution of percentage of TCRs co-clustered with COVID-19 references of query samples in each reference set configuration. x-axis displays the query sample true class labels.

- Fig S8: you show Spearman correlation. What are you correlating?

From TCR clustering data with N samples, we calculated the percentage of TCRs of each sample co-clustered with each of the other samples. We assigned the self-co-clustering percentage to be zero, to make all the vectors length N. The Spearman correlation matrix was calculated from the N-by-N co-clustering fraction matrix. We added this description to the legend of Fig S8 (now Fig S9).

- Fig 3: what if you choose something more difficult than leave one out? How does predictive performance change?

We randomly selected 40% of the reference samples as the training data, leaving the remaining 60% as test data. Training samples are labeled with “COVID-19”, “Cancer”, “MS” or “HC”. Each test data was co-clustered with all the training samples to calculate the fraction of TCRs clustered with each sample category. The fraction of “HC” is used to distinguish diseased vs healthy individuals. Other fractions are used to differentiate the 3 diseases, as described in the Methods section. We observed the same level of prediction accuracy as leave-one-out-validation. The result is as below:

Figure for review 10: Training accuracy estimated from 40% to 60% split of the training dataset, instead of leave-one-out-validation.

- Fig 4: “Since the query samples were derived from studies not included in the reference data, the high AUCs were not caused by unknown batch or cohort-specific effects, thus likely reflected the real predictability for the three diseases.” This statement is very unclear to me. Studies sequenced with different protocols could very well explain high predictive performance.

We agree with the reviewer that there are multiple TCR sequencing protocols available, such as Adaptive, 5/3 RACE, mPCR-based, etc, and comparing cohorts generated using different protocols may result in biased predictive performance. However, all the sample cohorts used in our study were sequenced with the same hsTCRB immunoSEQ protocol from Adaptive Biotechnology (<https://www.immunoseq.com/assays/>). Therefore, TCR sequencing protocol was not a concern. However, samples from the same study cohort may share similarities due to artifacts generated from tissue collection, sample storage, DNA extraction or other procedures. Due to these concerns, prediction accuracy derived from the same cohort may be overestimated. We explicitly mentioned this possibility in the manuscript, after observing the high AUCs in the training data in Fig 3b (now Fig 4b): “However, as most of the diseases were derived from only one study, it raised the concern that the predictability might be contributed by unknown batch effects.” Therefore, we conducted the experiment in Fig 4 (now Fig 5) to evaluate the performance where query samples were derived from different cohorts, which is able to rule out the same-cohort-artifacts.

- Please provide a reference for the following claim regarding repertoire classification: "The common limitation of these methods is their high computational cost that prevents from scaling up to large TCR-seq datasets"

Based on the data provided by in the DeepRC manuscript (0.0135s per update for 4 samples, with at least 10^5 updates), GIANA is over 100 times faster. Despite this fact, the computational time complexity of DeepRC or other MIL-based methods should still be affordable when applied to large samples (~ weeks). Therefore, we apologies for this confusion, and removed this sentence in the manuscript.

- For the statement "Interestingly, lung cancer and COVID-19 patients formed a separate cluster", any result that is challenging to justify can be viewed as interesting, but it should also be discussed what this might be indicating in terms of weaknesses of the approach.

Co-clustering of lung cancer and COVID-19 patients suggested an elevated sharing of TCRs among individuals bearing either disease. Our hypothesis for this elevation is the sharing of lung tissue-associated TCRs, which we provided further evidence to support (Fig S8). The major weakness of this approach, however, is that we cannot

definitely prove that the shared TCRs between the two cohorts are specific to lung antigens. This experiment is currently not feasible due to 1) lack of knowledge of shared lung-specific antigens in COVID-19 and lung cancer patients; 2) unknown alpha chain, which makes it impossible to produce full-length T cell receptors. We discussed this weakness in the revised manuscript.

Introduction

- Please cite <https://pubmed.ncbi.nlm.nih.gov/30247624/> and discuss how your paper is different or similar in measuring cluster accuracy

Please cite: <https://www.biorxiv.org/content/10.1101/2021.02.22.432291v1> and discuss how your paper differs

We cited both manuscripts in the revision. Following the reviewer's suggestion, we applied similar method in the first paper to define cluster purity and retention. We also modified the definitions of the measurements to be consistent with literature and to avoid any potential confusions. The second paper is a preprint and has not been peer reviewed. We tried to run clusTCR on our system, but got a fatal GPU error even though the use_gpu option was set False. This method also applies a numeric encoding strategy to project CDR3 sequences into high-dimensional space. GIANA is different in at least two aspects: 1) in addition to CDR3s, GIANA also consider TCR variable gene allele information to increase clustering accuracy; 2) GIANA performs Smith-Waterman alignment for all selected pre-cluster TCRs, to eliminate false-positive clustering due to variations in the encoding coordinate system. Conceptually, clusTCR is similar to GIANAsv with -e and -v options (non-exact mode and without variable genes).

- Please re-introduce the abbreviation of GIANA in the introduction and discuss to what extent the antigen-specific part of the name is relevant with respect to the clustering and prior literature

We changed the full name of GIANA into: Geometric Isometry based TCR AlignMent Algorithm, to avoid the confusion about antigen-specificity, and added it to the introduction.

Discussion

- "In summary, GIANA is a novel antigen-specific TCR clustering algorithm that is able to efficiently handle tens of millions of sequences." Again, to what extent is GIANA built with clustering antigen specificity in mind? Specifically, what do we learn about antigen-specific patterns using GIANA?

We agree with the reviewer that GIANA clustering was not dependent on the input of antigen-specific information, and removed the term "antigen-specific" in this sentence, and elsewhere in the manuscript, to lift the confusion.

- Can the authors discuss to what extent their method is dependent on the accuracy of the reference database as well as the metadata of the samples (age, sex, etc)? It is known these factors can influence the repertoire – was sex, age etc matched between HC, Covid and Cancer? If available can you provide the metadata in the supplementary? Similarly, was sequencing depth similar across the three immune states? To what extent is your method vulnerable to different size studies? Has the data of the studies in question been previously published? If so, cite them corresponding papers if not already done.

The performance of reference-based TCR repertoire classification depends on the similarity of the query sample and the reference samples. For a given query, if there exist a few reference samples showing high similarity, it is likely that the class label will be assigned to these samples. Considering the factors that could influence the repertoire, a high quality reference dataset should comprehensively include subjects of different age, gender, ethnicity, etc. The reference in our work covered a large age span for both COVID-19 patients (8 – 89 yrs) and healthy controls (1 – 74 yrs), including balanced number of male and female subjects. This wide coverage provides a high probability to find the matched reference sample(s) when an unknown TCR-seq sample is being queried. We have provided the metadata of age and gender information for the HC and COVID-19 cohorts (uploaded to Zenodo <http://doi.org/10.5281/zenodo.4698929>). Unfortunately, for most of the cancer cohorts,

detailed clinical information was not available.

As for sequencing depth, it is challenging to ensure that different datasets have the same library size, which could cause vulnerability in the downstream analysis if not properly handled. In fact, it is known that repertoire diversity measures are heavily confounded by sequencing depth (Laydon et al., 2015). Our solution is to use the CDR3 sequences of the top 10,000 most abundant TCR clones, and to avoid using the clonal frequency estimations. This solution was proven robust against the fluctuations with variable library sizes (Beshnova et al., 2020). In this work, we continued to apply this strategy to avoid the impact of this potential confounder.

All the data used in this work has been published. The PubMed IDs of each study are provided in the revision (Table S4-S5), and properly cited in the revised manuscript.

Minor

- Please add citations to graphics tools used for plotting to the Methods section
The authors use the pROC package but only cite R but not proc. <https://cran.r-project.org/web/packages/pROC/citation.html>. Please check if all other computational tools have been cited correctly.

We added citations for igraph and pROC packages in the Methods section.

Reviewer #3 (Expertise: Computational and statistical biology, cancer immunology):

In this manuscript, the authors described a novel method GIANA to identify TCRs targeting the same antigens. GIANA implemented a smart algorithm to convert each CDR3 into a vector space so the sequence similarity can be approximated by Euclidean distance. The transformation allowed a highly efficient method to pre-cluster similar CDR3s and reduced the overall computational cost for TCR clustering and query. The authors not only showed improved accuracy and speed of GIANA against other state-of-the-art methods, but also demonstrated several biological applications such as identifying COVID-19 patients from blood TCRs. GIANA has been tested successfully on our system smoothly and produced meaningful results. However, there are a few technical issues that should be addressed or clarified before this paper can be published on Nature Communication.

We highly appreciate the reviewer's professional and responsible attitude by testing our method in person. We are deeply encouraged to learn that our method was able to run smoothly and produced meaning results in this independent test. In this revision, we have followed all of the reviewer's suggestions to address the technical issues and lift potential confusions.

1. The Euclidean distance between the transformed vectors for CDR3s is critical for clustering. What is the default distance threshold GIANA adopted to obtain the pre-clusters, and why is this value chosen? Even though the authors showed a strong correlation between the Euclidean distance of the transformed vectors and Smith-Waterman alignment scores, it would be more accurate to quantify the Euclidean distance approximation. For example, when fixing a CDR3 and a large set of CDR3s with desirable Smith-Waterman score, what is the proportion of those CDR3 with Euclidean distance higher than the default threshold (-t)?

Furthermore, the authors only mentioned that the Euclidean distance approximation is less accurate with more than six consecutive mismatches. However, the same issue happens if there are only two mismatches that are six amino acids apart, which is more common in practice. The experiment above could also help explain such

cases.

We thank the reviewer for this highly productive suggestion. The default value for isometric distance (-t option) is 10. Here we provide an justification of this value, by performing the analysis proposed by the reviewer using a previously described dataset of 10,000 cancer-associated TCRs of different lengths (Zhang et al., 2020, Clin Can Res). First, TCRs were divided into different length groups, and within each group, Smith-Waterman alignment scores were calculated using the BLOSUM62 substitution matrix for each pair of sequences. We then calculated the Euclidean distance after isometric encoding of each CDR3. We choose -S 3.5 (now default is 3.6) as the cutoff for SW alignment score. This translates to a raw alignment score of $3.5 \times (L - 5)$, where L is the sequence length. As shown in the figure below, all the sequence pairs with alignment score greater than $3.5 \times (L - 5)$ have isometric distances smaller than 10 (the default cutoff for -t option):

Figure for review 11: Sufficiently large isometric distance cutoff to include highly similar TCR pairs.

For 6-th order cyclic group, if two mismatches are exactly six amino acids apart, the transformation matrix becomes the identity matrix, which results in non-zero cross terms in the distance calculation and causes uncertainty in TCR clustering (please see the revised Methods section “*Case 4: Non-consecutive Mismatches*” for details). However, the above results show that current parameterization of GIANA is able to capture all the qualifying TCR pairs even in this scenario. We have added the above discussion in the revised manuscript.

2. GLIPH2 may assign one CDR3 to multiple clusters, and some of the large clusters are pretty aggressive, which combines multiple more fine-grained clusters. Therefore, GLIPH2 has a low number of “pure clusters” and “pure TCR” when considering all clusters. Could the authors explain how to handle such cases when comparing the sensitivity and precision of GLIPH2 and GIANA?

The reviewer is correct that when clustering multiple similar TCR sequences, there is a ‘small-world’ effect, meaning that TCRs with small hamming distances connect with each other, forming a large network with dissimilar sequences which are usually not informative to antigen-specific TCR inference. Small-world effect aggravates with larger amount of TCRs. GLIPH2 attempted to solve this problem by cutting the large network into smaller cliques, allowing for repeated occurrence of the same sequence in multiple clusters. This procedure, in theory, is expected to increase clustering purity in GLIPH2 output. Therefore, we treated each of the GLIPH2

cluster as independent clusters when calculating clustering purity. When computing the fraction of pure TCRs (sensitivity), we pooled TCRs from all the pure clusters, and took the unique ones to avoid repetition. Both measures (clustering purity or precision, and sensitivity) might favor smaller clusters. Therefore, in the revision, we followed reviewer's suggestion (below) to add Normalized Mutual Information as a 3rd measure to evaluate the all methods in this work.

3. The measure of precision and sensitivity using “pure cluster” may not be sufficient for the evaluations. For example, one method could call many small clusters, hence having a higher chance to obtain pure clusters and achieving higher performance. Measures like adjusted rand index or normalized mutual information could be more useful to compare the true TCR clusters and the predicted TCR clusters.

We followed the reviewer's suggestion and calculated the normalized mutual information (NMI) between TCR clusters called from various methods and the true epitope specificity labels. We observed similar levels of NMI for all four methods, where GLIPH2 remains to be the lowest (figure below). This measure describes the clustering results from a different perspective, which has been added to our revised manuscript as new Figure 2c:

Figure for review 12: Normalized mutual information for each method, by comparing TCR clusters with annotated antigen specificities.

4. For the GIANA algorithm, the authors used G_2XG_3 to transform the amino acid representations on different positions to orthogonal spaces. Would the more naive transformation like shifting each vector to the next r position work?

By “shifting each vector to the next r position”, we believe that the reviewer is describing the same approach as the “stacked vector representation”, or GIANAsv method in our manuscript. GIANAsv was compared to other methods in our work, and is slower than GIANA by a factor of 2.2 (new Fig 2a). It also consumes higher amount of memory due to larger encoding dimensionality.

5. In the abstract, the authors claimed GIANA could process 100 billion sequence comparisons within 3 minutes. However, this is misleading. With pre-cluster and k-mer guidance, GIANA conducted a much smaller number of sequence comparisons when querying 10^4 TCRs against 10^7 reference sequences. So the computational load is not equal to 100 billion pairwise comparisons.

We agree with the reviewer that the actual computational load in the query is not equal to 100 billion comparisons. We revised this sentence in the abstract into: “GIANA also allows ultrafast query of large reference cohorts, *which is able to compare 10^4 TCRs with 10^7 reference sequences within 3 mins.*”

Reviewers' Comments:

Reviewer #1:

Remarks to the Author:

The authors have satisfactorily answered all my previous questions. The only minor issue is that there are numerous typos in the manuscript and the authors may want to carefully proofread the manuscript again and fix them.

For example:

In the Methods Section, the 1st sentence: all TCR repertoire sequence samples were access[ed] via ...

Reviewer #2:

Remarks to the Author:

The authors have addressed all my concerns.

One minor remark. Widrich et al has in fact been peer-reviewed (NeurIPS, 2000):

https://proceedings.neurips.cc/paper_files/paper/2020/hash/da4902cb0bc38210839714ebdcf0efc3-Abstract.html

Reviewer #3:

Remarks to the Author:

The authors have done an impressive job of addressing the concerns. The revision has substantially improved the quality of the manuscript, especially in the algorithm explanation. However, there are two more concerns about the newly added Normalized Mutual Information (NMI) evaluation.

1. When computing NMI, each CDR3 should be assigned to one cluster id. Since in GLIPH2's result, one CDR3 could show up in multiple clusters, could the authors explain how the final cluster assignment was determined for GLIPH2's results? If the authors assigned a unique cluster id for each CDR3 in the NMI experiment, what are the purity cluster fraction and retention in this case?

2. Since the NMI score should be between 0 and 1, why the values from the y axis in Figure 2c are above 1?

Reviewer #4:

None

Reviewer #1 (Remarks to the Author):

The authors have satisfactorily answered all my previous questions. The only minor issue is that there are numerous typos in the manuscript and the authors may want to carefully proofread the manuscript again and fix them.

For example:

In the Methods Section, the 1st sentence: all TCR repertoire sequence samples were access[ed] via ...

Following reviewer's suggestion, we have extensively revised the manuscript to fix the typos.

Reviewer #2 (Remarks to the Author):

The authors have addressed all my concerns.

One minor remark. Widrich et al has in fact been peer-reviewed (NeuRIPS, 2000): https://proceedings.neurips.cc//paper_files/paper/2020/hash/da4902cb0bc38210839714ebdcf0efc3-Abstract.html

We thank the reviewer for providing this information for our future reference.

Reviewer #3 (Remarks to the Author):

The authors have done an impressive job of addressing the concerns. The revision has substantially improved the quality of the manuscript, especially in the algorithm explanation. However, there are two more concerns about the newly added Normalized Mutual Information (NMI) evaluation.

1. When computing NMI, each CDR3 should be assigned to one cluster id. Since in GLIPH2's result, one CDR3 could show up in multiple clusters, could the authors explain how the final cluster assignment was determined for GLIPH2's results? If the authors assigned a unique cluster id for each CDR3 in the NMI experiment, what are the purity cluster fraction and retention in this case?

The reviewer is correct that in GLIPH2, some CDR3s may show up in multiple clusters. This becomes a sticky point when calculating the NMIs if taking all the clusters together. Our solution is, we treated the different GLIPH2 clusters as if they were independent. To estimate NMI, we calculated the NMI for each cluster and took the averaged value. This is an unbiased estimator because NMIs from all clusters can be assumed to be identically distributed, although they are not independent due to shared CDR3s. Based on the law of large numbers (WLLN), when the sample size approaches to infinity, the averaged value of the random variables will approach to their expectation, with bounded covariance of the random variables. A proof using the Chebyshev's inequality is available at this link:

<https://math.stackexchange.com/questions/245327/weak-law-of-large-numbers-for-dependent-random-variables-with-bounded-covariance>.

Thus, the mean estimator is unbiased. In fact, the correlation between different clusters due to shared CDR3s would affect the estimation of the second moment, which is the variance of NMI, but it is not a concern of our analysis. As we did not seek to assign unique cluster IDs to each CDR3, the same argument is applied to estimate cluster purity and retention. We have added the above description in the Methods section.

2. Since the NMI score should be between 0 and 1, why the values from the y axis in Figure 2c are above 1?

We double checked our code and found that we were calculating the mutual information, instead of the normalized mutual information. We fixed this error and replaced the plot of Figure 2c. The new result does not change any of our conclusions.